# Molecular Phylogenetics and Estimation of Evolutionary Divergence and Biogeography of the Family Cordycipitaceae (Ascomycota, Hypocreales)

**DOI:** 10.3390/jof11010028

**Published:** 2025-01-02

**Authors:** Huili Pu, Jiao Yang, Nemat O. Keyhani, Lixia Yang, Minhai Zheng, Chenghao Qiu, Yuchen Mao, Junya Shang, Yongsheng Lin, Chengjie Xiong, Longbing Lin, Pengyu Lai, Yanbin Huang, Xin Yuan, Huiling Liang, Longfei Fan, Xiaoli Ma, Chunjin Qiu, Junzhi Qiu

**Affiliations:** 1State Key Laboratory of Ecological Pest Control for Fujian and Taiwan Crops, College of Life Sciences, Fujian Agriculture and Forestry University, Fuzhou 350002, China; hdpuhuili@163.com (H.P.); jiaoyang0368@163.com (J.Y.); finy1015@163.com (L.Y.); zhengmh0928@163.com (M.Z.); maoyuchen301@163.com (Y.M.); 15750811500@163.com (J.S.); linyongsheng0909@163.com (Y.L.); 18609200812@163.com (C.X.); llb_77@126.com (L.L.); ndpengyu@163.com (P.L.); 2Department of Biological Sciences, University of Illinois, Chicago, IL 60607, USA; keyhani@uic.edu; 3XiangYa School of Public Health, Central South University, Changsha 410013, China; chenghaoqiu2023@126.com; 4Bureau of Fujian Junzifeng National Nature Reserve, Sanming 365200, China; mxhyb@163.com; 5Bureau of Fujian Longqishan National Nature Reserve, Sanming 353300, China; 18759868636@163.com; 6Guangxi Institute of Botany, Chinese Academy of Sciences, Guilin 541006, China; llhl@gxib.cn; 7College of Plant Protection, Gansu Agricultural University, Lanzhou 730070, China; fanlf@gsau.edu.cn; 8College of Life Science and Technology, Xinjiang University, Urumqi 830046, China; xlm87@xju.edu.cn; 9Putian Institute of Agricultural Sciences, Putian 351106, China

**Keywords:** Cordycipitacea, ancestral state reconstruction, molecular clock, fungal morphology, multi-locus phylogeny, fungal taxonomy

## Abstract

The Cordycipitaceae family of insecticidal fungi is widely distributed in nature, is the most complex in the order Hypocreales (Ascomycota), with members displaying a diversity of morphological characteristics and insect host ranges. Based on Bayesian evolutionary analysis of five genomic loci(the small subunit of ribosomal RNA (SSU) gene, the large subunit of ribosomal RNA (LSU) gene, the translation elongation factor 1-α (*tef1-α*) gene, the largest subunit of RNA polymerase II (*rpb1*), and the second largest subunit of RNA polymerase II (*rpb2*), we inferred the divergence times for members of the Cordycipitaceae, improving the internal phylogeny of this fungal family. Molecular clock analyses indicate that the ancestor of *Akanthomyces sensu lato* occurred in the Paleogene period (34.57 Mya, 95% HPD: 31.41–37.67 Mya), and that most species appeared in the Neogene period. The historical biogeography of *Akanthomyces sensu lato* was reconstructed using reconstructing ancestral state in phylogenies (RASP) analysis, indicating that it most likely originated in Asia. Combined morphological characterization and phylogenetic analyses were used to identify and taxonomically place five species within Cordycipitaceae. These include the following: (i) two new species, namely *Akanthomyces baishanensis* sp. nov. and *Samsoniella sanmingense* sp. nov., (ii) a new record species isolated from infected Lepidopteran host, *Blackwellomyces lateris*, (iii) a new record species in the genus *Niveomyces*, with *sporothrix*-like asexual morphs, namely *N. multisynnematus*, isolated from dipteran insects (flies), and (iv) a known species of the (hyper-) mycoparasite, *Liangia sinensis*, isolated from the fungus *Ophiocordyceps globiceps* (Ophiocordycipitaceae) growing on a dipteran host. Our data provide a significant addition to the diversity, ecology, and evolutionary aspects of the Cordycipitaceae.

## 1. Introduction

The Cordycipitaceae family of parasitic fungi is one of the most complex and diverse taxa in the order Hypocreales (Ascomycota). These fungi are well known pathogens of a wide range of insects and are often morphologically characterized by pale to brightly colored stromata with entirely or partly immersed or superficial perithecia [1,2]. In addition, three types of ascospores have been described for these fungi, namely disarticulating, intact, and bola [3]. The taxonomy of the family Cordycipitaceae is continually being updated with the recognition and incorporation of new genera, e.g., *Arachnidicola* [4], *Bhushaniella* [5], *Corniculantispora* [4], *Jenniferia* [6], *Pseudoniveomyces* [7], and *Niveomyces* [7], leading to revisions to phylogenetic classification. More than 100 species have been described thus far, which display diverse morphological characteristics, insect host ranges, and trophic levels, e.g., leaf litter, bark, decaying logs, plant stems, and soil, that match host environmental niches, including those found underground. Hosts for these fungi range across most insect orders (e.g., Hemiptera, Hymenoptera, Lepidoptera, Coleoptera, Diptera, and Orthoptera) and even across Arthropoda to spiders and mites [2,8,9]. Some species of Cordycipitaceae are highly prized, with significant economic value, and are harvested for use in (traditional) medicinal and health-related products. Metabolites from these fungi, e.g., cordycepin and other compounds from *Cordyceps militaris*, have been shown to possess a range of antiviral, antibacterial, antioxidant, antiproliferative, and/or immunomodulatory pharmacological activities [10,11,12]. In addition to medicinal value, some species have been commercialized for use in pest control applications relevant to agriculture, forestry, and human health, e.g., species within the *Beauveria*, *Metarhizium*, *Isaria*, and *Cordyceps* genera [13,14,15,16].

*Samsoniella* is a genus of invertebratepathogenic fungi within the Cordycipitaceae. Sexual morphs of *Samsoniella* are known for their bright red–orange stromata and longitudinal ditch on the side of the fertile part [2,17]. However, in some instances, members assigned to the genus are similar to *Akanthomyces sensu lato* (*s.l*.) speciesin micro-morphological characteristics [18]. Species within *Samsoniella* mainly parasitize Lepidoptera larvae or pupae (butterflies and moths) in soil or leaf litter, and can be developed into biological control agents against a variety of pests; however, their fruiting bodies are also highly prized as having important (traditional) medicinal value, especially in Asian cultures [19]. To clarify the phylogenetic diversity of *Isaria*-like species, Mongkolsamrit et al., based on molecular phylogenetic analysis of five different loci (ITS, 28S rDNA, *tef1-α*, *rpb1*, and *rpb2*), assigned *Samsoniella* to a group of members possessing *Cordyceps*-like sexual morphs and *Isaria*-like asexual morphs [17]. Subsequently, recessive intraspecific diversity among different isolates and genotypes of *Isaria*-like species has been characterized, leading to revisions in the systematic positions of many *Isaria*-like species within the Cordycipitaceae family [2,20]. For example, *Paecilomyces hepiali* (previously classified within *Isaria*-like species) has been transferred into *Samsoniella*, and renamed *S. hepiali* [2]. In addition, nine new species within *Samsoniella*, namely *S. alpina*, *S. antleroides*, *S. cardinalis*, *S. cristata*, *S. lanmaoa*, *S. kunmingensis*, *S. ramosa*, *S. tortricidae*, and *S. yunnanensis*, were described. New species of *Samsoniella* continue to be found in forest habitats and karst ecological environments in different provinces of China [9,21].

The genus *Akanthomyces* was established by Lebert in 1858, with *Akanthomyces aculeatus* as the type species. Kepler et al. [22] conducted a molecular phylogenetic study of Cordycipitaceae using multi-gene sequence data and revising the systematic position of some taxa of the family (such as transferring *Isaria* species into *Cordyceps*), and suggested that nine genera should be included in Cordycipitaceae. Moreover, they found that some species of the genus *Akanthomyces* and *Cordyceps* could not be included in the family, with a new genus, *Hevansia* (Luangsaard, Hywel-Jones and Spatafora) established to accommodate excluded species. Recently, based on multi-locus phylogenetic analysis of LSU, *tef1-α*, *rpb1*, and *rpb2* sequences, Khonsanit et al. [4] proposed that *Akanthomyces s.l.* should consist of four monophyletic groups, including *Akanthomyces*, *Arachnidicola*, *Lecanicillium*, and *Kanoksria*. To accommodate *Akanthomyces* species that exhibit *Lecanicillium*-like asexual morphs, it is recommended to resurrect the *Lecanicillium* genus.

The genus *Liangia* often displays *Lecanicillium*-like asexual characteristics. It was isolated by Yu et al. and currently contains only one species, *Liangia sinensis* [2]. Recently, the genus *Niveomyces* (Cordycipitaceae, Hypocreales) was isolated by Araújo, J. P. M. and de Bekker, C. [23] with *N. coronatus* as the type species. Members of the *Niveomyces* directly produce multiple synnemata on host tissue or subiculum, and the host is usually covered with a white to cream-colored mycelium [7]. The asexual morph is characterized by hyaline, smooth oval to cylindrical conidia with rounded ends, and scattered, denticulate conidiogenous cells arising directly from the hyphae or from synnemata [7,23]. Currently, *Niveomyces* consists of six species: *N. albus*, *N. coronatus*, *N. formicidarum*, *N. hirsutellae*, *N. insectorum*, and *N. multisynnematus*.

Despite their ubiquity, important aspects of the species diversity, taxonomic status, and phylogenetic relationships within Cordycipitaceae remain to be clarified. During an exploration aimed at delineating the diversity and geographical distribution of entomopathogenic fungi in China, a series of intriguing specimens were collected in the Fujian and Jilin Provinces, China. These isolates were characterized via molecular sequencing of five genomic loci, and phylogenetic reconstrcution was used to clarify their taxonomic placement within Cordycipitaceae. With these data, combined with morphological characterization, we report the description of two new species, two new record species, and one known species, which are illustrated and described in detail. Concurrently, we utilized *Paleoophiocordyceps coccophagus* (G. H. Sung, Poinar and Spatafora) as a fossil calibration for molecular clock analysis to infer the divergence times of members of Cordycipitaceae, increasing our understanding of the phylogeny, origin, and evolutionary history of this family.

## 2. Materials and Methods

### 2.1. Sampling, Fungal Isolation, and Morphology

Entomopathogenic fungi were surveyed in Fujian and Jilin Province, China. Purification of single spore colonies of putative new species was isolated as described [7]. Images of the fresh specimens were captured using a Canon (Tokyo, Japan) EOS 6D Mark II camera. Briefly, fungal cultures were isolated from fresh conidia or mycelia of insect cadavers using sterile needles and staking onto potato dextrose agar (PDA). Plates were incubated at 25 °C and 60% relative humidity (RH) for 5–14 d. Nine fungal specimens were isolated in total.

Purified fungal cultures were transferred to PDA plates and incubated for 14 d. A digital camera was used to take photos of the colonies for measurement of colony diameters. A lactophenol cotton blue solution was used as the medium for microscope observation, and the fresh structure of the pure cultures was put into the solution for microphotography [2]. The micro-morphological structure of the purified isolates was observed using a Nikon Ni-U compound microscope and via scanning electron microscopy (SEM, Nikon Ni-U; HITACHI SU3500, Tokyo, Japan). Measurements on images were performed using Digimizer 5.4.4 software. All cultures were stored in 10% sterilized glycerin and sterile water at 4 °C for future studies. The dried specimens and living cultures were deposited in the Herbarium Mycologicum Academiae Sinicae, Institute of Microbiology, Chinese Academy of Sciences, Beijing, China (HMAS), and the China General Microbiological Culture Collection Center (CGMCC).

### 2.2. DNA Extraction and PCR Amplification

Total genomic DNA was extracted from fungal colonies grown on PDA plates using the Fungal DNA Mini Kit (OMEGA–D3390, Feiyang Biological Engineering Corporation, Guangzhou, China) following the manufacturer’s recommendations. Nucleotide sequences were obtained from five genomic loci, including the small and large subunits of nuclear encoded ribosomal DNA genes (SSU and LSU), the translation elongation factor 1α (*tef1-α*), and the largest and second largest subunits of RNA polymerase II (*rpb1* and *rpb2*). The following primer sets were utilized to amplify various genetic markers via polymerase chain reaction (PCR): nrSSU-CoF/nrSSU-CoR for SSU [24], LR0R/LR5 for LSU [25,26], EF1α-EF/EF1α-ER for *tef1-α* [27], RPB1-Ac/RPB1-Cr for *rpb1* [28,29], and fRPB2-5F/fRPB2-7cR for *rpb2* [30], as previously described [1]. Primer sequences and annealing temperatures of different primers were noted in Table 1. Polymerase chain reaction (PCR) was performed in 25 μL volumes consisting of 12.5 µL of 2 × Rapid Taq Master Mix (Vazyme, Nanjing, China), 1 µL of each forward and reverse primer (10 µM) (Sangon, Shanghai, China), 1 µL of genomic DNA, and 9.5 µL of sterilized distilled water. The PCR amplicons were sent to Tsingke Biotech Co., Ltd. (Fuzhou, China) for purification and Sanger sequencing. New sequences generated in this study have been deposited in GenBank (www.ncbi.nlm.nih.gov/genbank/, accessed on 8 January 2024).

### 2.3. Sequence Alignment and Phylogenetic Analyses

Reference sequences for the phylogenetic analysis were downloaded from NCBI (National Center for Biotechnology Information), based on BLAST search results in NCBI and in conjunction with the relevant literature (Table 2) [2,7,23,31,32,33]. Each locus was individually aligned using the MAFFT v.7.11 online tool, followed by trimming the consensus sequences from the different primers in MEGA 7.0 software. The concatenated aligned dataset was used for multi-locus phylogenetic analysis using maximum likelihood (ML) and Bayesian inference (BI) methods. The ML was run on IQtree 1.6.8 [34], available on the Phylosuite software v1.2.3 (https://dongzhang0725.github.io/, accessed on 3 September 2024), with the model being automatically selected and bootstrap iterations set to 1000.BI analysis was performed with MrBayes 3.2.6 [35], and the best evolutionary model was decided under the Akaike Information Criterion (AIC) via PartitionFinder2, in which the first 1/4 of the trees were discarded as burn-in trees, and four parallel Markov Chain Monte Carlo chains sampled every 100 steps for 2,000,000 generations, until the average standard deviation reached 0.01. The phylogenetic trees were visualized using the Interactive Tree of Life (iTOL, https://itol.embl.de, accessed on 6 July 2024) online tool [36], and edited with Adobe Illustrator CS 6.0 (Adobe Systems Inc., San Jose, CA, USA).

### 2.4. Divergence Time Estimation

Based on the SSU, LSU, *tef-1α*, *rpb1*, and *rpb2* concatenated sequence dataset derived from 119 specimens, the divergence times of *Akanthomyces s.l*. and other members of the Cordycipitaceae species were inferred, as described [37,38]. Each partition was divided by MrModeltest v 2.3, and the GTR + G + I substitution model was selected as the best-fit model. A relaxed clock log normal model was used for BEAST [39,40,41]. In divergence time estimation, a fungal parasite *P. cocophagus* from a Cretaceous period (99–105 Mya) scale insect was used as the fossil calibration, and the fossil node calibration was calibrated using the gamma prior distribution [37,42]. The XML (Extensible Markup Language) file generated by BEAUti (version 2) was then executed in BEAST. A relaxed clock log normal model was used for BEAST. Four independent MCMC chains of 100 million generations were conducted and logging parameters were sampled every 10,000 generations. The resulting log file was checked for convergence using the program Tracer v.1.6 (http://tree.bio.ed.ac.uk/software/tracer/, accessed on 19 August 2024; ESS ≥ 200 was considered convergence). When ESS ≥ 200, a maximum clade tree was created in TreeAnnotator v. 2.6.7, where the first 10% of trees were discarded as burn-in trees. The final tree was edited using the tvBOT online program (https://www.chiplot.online/tvbot.html, accessed on 30 October 2024) [43].

### 2.5. Inferring Historical Biogeography

We used the Reconstruct Ancestral State in Phylogenies (RASPv.4.3) software package to infer the historical biogeography of *Akanthomyces sensu lato*, by reconstructing the ancestral geographical distribution on the phylogenetic tree [44,45]. In the reconstruction work, the Bayesian Binary Markov chain Monte Carlo (BBM) method provided in RASP (version 4.3) was used for analysis with 10 million generations [46]. The first 10% of trees were discarded as burn-in trees and the other parameters remained the default settings. Statistically, the geographic distributions of *Akanthomyces s.l*. were identified in five areas: (A) Asia, (B) North America, (C) South America, (D) Europe, and (E) Africa.

## 3. Results

### 3.1. Phylogenetic Analyses

The molecular analyses and morphological comparisons presented herein indicate that the Fujian and Jilin Province specimens collected belong to Cordycipitaceae and group within the *Liangia*, *Niveomyces*, and *Samsoniella* genera. Two new species (*Akanthomyces baishanensis* sp. nov. and *Samsoniella sanmingense* sp. nov.), two new record species (*Blackwellomyces lateris* and *Niveomyces multisynnematus*) and one known species (*Liangia sinensis*) are described, and their phylogenetic placements determined below (Table 2, Figure 1).

Phylogenetic analyses were based on the concatenated nucleotide sequence dataset derived from the SSU, LSU, *tef–1α*, *rpb1*, and *rpb2* gene loci, and were examined across 29 taxa representing 235 sequences within Cordycipitaceae, which included *Trichoderma* (*T. deliquescens* ATCC 208838 and *T. stercorarium* ATCC 62321) as the outgroup as detailed in the Materials and Methods section (Table 2, Figure 1). The combined dataset consisted of 4473 characters (i.e., SSU: 1–1002 bp, LSU: 1003–1734 bp, *tef1-α*: 1735–2768 bp, *rpb1*: 2769–3455 bp, and *rpb2*: 3456–4473 bp), 2593 of whichwere constant, and 2450 distinct patterns, 355 singleton sites, and 1525 parsimony-informative sites were identified. The best substitution model for BI analysis included SYM + I + G4 for SSU, SYM + G4 for *rpb2*,and GTR+F+I+G4 for LSU, *tef1-α*, and *rpb1*, resulting in an average standard deviation of split frequencies = 0.007254. For ML and BI phylogenetic analyses, the inferred topologies were largely consistent across most branches, with *Ak. baishanensis*, *B. lateris*, *L. sinensis*, *N. multisynnematus*, and *S. sanmingense* found to be distributed within distinct clades with strong support.

### 3.2. Dating and Evolutionary Analyses of Akanthomyces s.l. in Cordycipitaceae

To explore the evolution, origin, and evolutionary history of the internal systematics of *Akanthomyces s.l.*, all species of *Akanthomyces s.l*. and other members of Cordycipitaceae were selected for study based on previous phylogenetic research. These analyses confirmed that the Cordycipitaceae diverged from the early Cretaceous period [37,38] (Figure 2). These analyses further indicate that the divergence time estimate shows that most species within *Akanthomyces s.l.* appeared later than other genera of Cordycipitaceae, with a mean stem age of 34.57 Mya (95% highest posterior density (HPD) of 31.41–37.67 Mya, 1.0 PP), and a mean crown age of 29.32 Mya (95% HPD of 26.68–31.97 Mya, 1.0 PP), and that in Cordycipitaceae, the initial diversification of *Akanthomyces s.l.* occurred during the Paleogene period (Figure 2, Table 3). The subsequent divergence of most species within *Akanthomyces s.l.* occurred mainly during the Neogene period.

### 3.3. The Historical Biogeography of Akanthomyces s.l.

Historical biogeographic scenarios for *Akanthomyces s.l.* were inferred using RASP, as detailed in the Methods section (Figure 3). The results of BBM analysis indicate that *Akanthomyces s.l.* has a complex biogeographical history, with at least 13 dispersal events and four vicariance events determining the current distribution of the genus. This analysis supports the finding that Asia has the highest probability of being the ancestral area of *Akanthomyces s.l.*, with 31 species currently found in Asia, three in North America, four each in South America and Europe, and one in Africa, suggesting that Asia is still the center of *Akanthomyces s.l.* diversification.

### 3.4. Taxonomy

#### 3.4.1. *Akanthomyces baishanensis* sp. nov., Pu, H.L. and Qiu, J.Z., Figure 4c

MycoBank Number: MB852386

Etymology: Named after the location where the type specimen was found, Baishan City, Jilin Province.

**Figure 4 jof-11-00028-f004:**
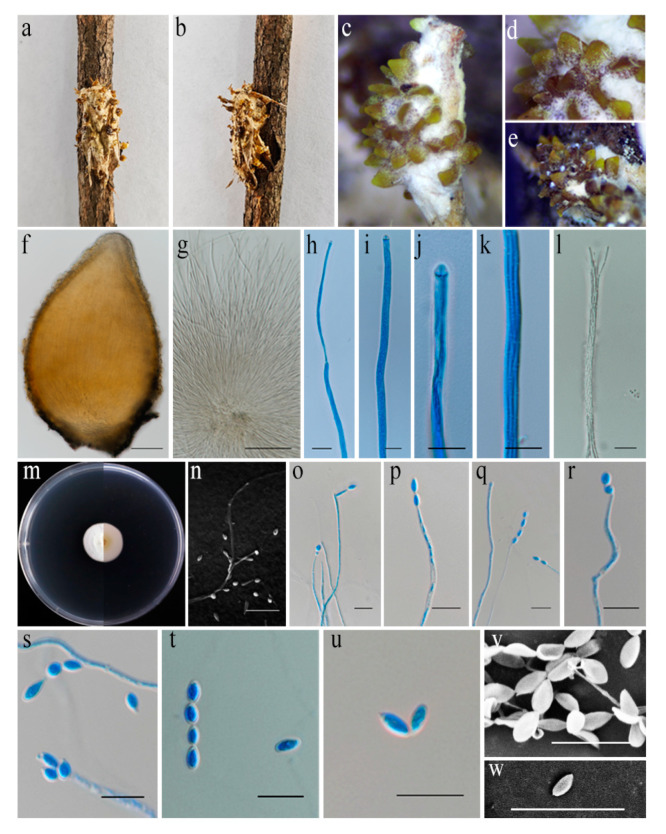
*Akanthomyces baishanensis* (HMAS 298137). (**a**,**b**) Fungus on adult moth. (**c**–**e**) Close-up of stromata. (**f**) Perithecia. (**g**,**h**) Ascus. (**i**,**j**) Ascus tip. (**k**,**l**) Part of asci showing ascospores. (**m**) Upper and reverse view of colony after incubation for 14 days on PDA. (**n**–**s**) Conidiogenous cells (conidiophores and phialides). (**t**–**w**) Conidia. Scale bars: (**f**,**g**) 100 µm; (**h**–**l**,**n**–**u**) 10 µm; (**v**–**w**) 5 µm.

Holotype: China, Jilin Province, Baishan City, 127°56′57.6″ E, 42°32′28.7″ N, on adult moth (Noctuidae, Lepidoptera), 4 September 2023, holotype HMAS 298137; ex-holotype living culture CGMCC3.25673.

Description: Adult moth attached to a plant stem partially covered by white to cream mycelium. Stromata arising from host body and wing veins, non-stipitate. Perithecia superficial, orange to light brown, growing directly from mycelium that covers the host body, ovoid, 573–658 × 293–479 µm. Asci cylindrical and hyaline, up to 150–229 × 4.2–5.0 μm, with a hemispheric apical cap of 3.7–4.7 × 3.8–4.4 µm. Ascospores hyaline, filiform, and multiseptate, 5.3–13.9 × 1.3–1.5 μm. Asexual morph not observed.

Culture characteristics: Colonies on PDA slow growing, up to 19–22 mm diameter in 14 days at 25 °C, growth rate 1.4–1.6 mm/day, white and cottony, growing in a circular pattern with entire margins, and closely appressed to the agar surface, reverse pale yellow, and turns white outward. Conidiophores mononematous hyaline and smooth-walled. Phialides hyaline, smooth, and consisting of a cylindrical, slightly inflated base, tapering to a thin neck, 8.7–26.5 × 1.7–2.7 μm, mean = 17 × 2.3 μm. Conidia in chains, smooth, hyaline, and fusiform to ellipsoidal, exists at the apex of phialides, 3.8–5.3 × 2.4–3.6 μm, mean ± SD = 4.7 ± 0.4 × 3.0 ± 0.3 μm, L/W ratio = 1.6, n = 30.

Material examined: China, Jilin Province, Baishan City, 127°56′57.6″ E, 42°32′28.7″ N, on adult moth (Noctuidae, Lepidoptera), 4 September 2023, paratype HMAS 298138; ex-paratype living culture CGMCC3.25674.

Notes: *Akanthomyces baishanensis* was easily identified as belonging to *Akanthomyces* based on the phylogenetic analyses. The new species *Ak. baishanensis* forms an independent clade and was phylogenetically distinct from *Ak. phariformis*, with strong support value (100% ML/1 PP, Figure 1). The nucleotide comparison of LSU sequences of *Ak. phariformis* revealed 6 bp (6/811 bp, 0.7%) nucleotide differences. The nucleotide comparison of *tef1-α* sequences of *Ak. phariformis* revealed 18 bp (18/900 bp, 2.0%) nucleotide differences. The nucleotide comparison of *rpb2* sequences of *Ak. phariformis* revealed 22 bp (22/822 bp, 2.7%) nucleotide differences. In morphology, this species differs from *Ak. phariformis* by producing crowded superficial perithecia at the host body and wing veins, while *Ak. phariformis* produces perithecia at the apex of synnemata. The perithecia of *Ak. baishanensis* are longer and wider than those of *Ak. phariformis* (573–658 × 293–479 vs. 360–550 × 200–260 μm). Ecologically, *Ak. baishanensis* is found in northern China (temperate), while *Ak. phariformis* is found in northern Thailand (tropical). In addition, we observed the conidia and reproductive structures of *Ak. baishanensis* on PDA, while *Ak. phariformis* has not been described thus far. Therefore, we describe this fungus as a new species.

#### 3.4.2. *Samsoniella sanmingense* sp. nov., Pu, H.L. and Qiu, J.Z. Figure 5

MycoBank Number: MB852399

Etymology: Named after the location where the type specimen was found, Sanming City, Fujian Province.

**Figure 5 jof-11-00028-f005:**
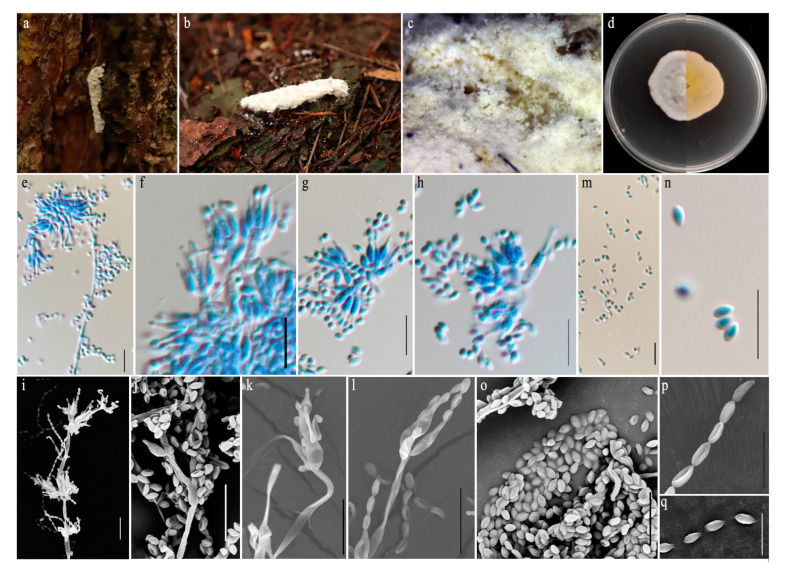
*Samsoniella sanmingense* (HMAS 352694). (**a**,**b**) Larva of Lepidoptera infected by *S. sanmingense.* (**c**) Close-up of synnemata. (**d**) Upper and reverse view of colony after incubation for 14 days on PDA. (**e**–**l**) Conidiogenous cells (conidiophores and phialides). (**m**–**q**) Conidia. Scale bars: (**e**–**m**,**o**) 10 µm; (**n**,**p**,**q**) 5 µm.

Holotype: China, Fujian Province, Sanming City, Mingxi County, Xiafang Town, Zhushe Village, 117°1′59.6″ E,26°33′53.6″ N, on the larvae of Lepidoptera buried in soil, 26 April 2023, holotype HMAS 352694; ex-holotype living culture CGMCC3.25662.

Description: Host covered by white to cream-colored and cottony mycelium. Conidiophores produced on prostrate aerial hyphae, smooth-walled, with a single phialide or whorls of two to five phialides, or *Verticillium*-like from hyphae directly, 4.9–10.9 × 2.4–3.1 μm, mean = 6.7 × 2.7 μm. Phialides smooth-walled, with a cylindrical basal portion, tapering gradually or abruptly toward the apex into a distinct neck, 4.5–11.4 × 1.9–2.6 μm, mean = 7.1 × 2.2 μm. Conidia in chains, hyaline, oblong, elliptical, or fusiform, 2.4–3.7 × 1.8–2.6 μm, mean ± SD = 3.1 ± 0.3 × 2.2 ± 0.2 μm, L/W ratio = 1.4, n = 30. Sexual morph not observed.

Culture characteristics: Colonies on PDA moderately fast-growing, 32–34 mm diameter in 14 days at 25 °C, growth rate 2.3–2.4 mm/day, white and cottony, floccose, consisting of a basal felt with high mycelium density, and reverse pale yellow.

Material examined: Fujian Province, Sanming City, Junzifeng National Nature Reserve, 117°2′1.4″ E, 26°33′53.6″ N, on the pupae of Lepidoptera clinging to fallen leaves, 26 April 2023, paratype HMAS 352693; ex-paratype living culture CGMCC3.25661.

Notes: Based on the sequences of the characterized SSU, LSU, *tef–1α*, *rpb1*, and *rpb2* genes, phylogenetic analyses revealed the isolate as belonging to the *Samsoniella* genus. However, *S. sanmingense* formed a separate subclade distinct from the other species within *Samsoniella*, with moderate 98% ML bootstrap support and 1 posterior probability (Figure 1).The pairwise dissimilarities of *tef1-α* sequences show 14 bp difference within 924 bp between *S. sanmingense* and *S. tiankengensis*. Our morphological studies show that *S. sanmingense* has different phialides and conidial strcutures from *S. tiankengensis*. The conidia of *S. sanmingense* are larger than those of *S. tiankengensis* (2.4–3.7 × 1.8–2.6 µm vs. 2.3–2.8 × 1.6–1.8 µm), and the phialides are also broader than those of *S. tiankengensis* (4.5–11.4 × 1.9–2.6 µm vs. 5.4–10.4 × 1.3–2.2 µm). Therefore, we describe this fungus as a new species.

#### 3.4.3. *Blackwellomyces lateris* Xiao, Y.P.; Wen, T.C.; Hyde, K.D. Fungal Diversity **2019**, *96*, 1–242 [47]; Figure 6

Description: See Hyde et al. [47]. Host covered by white to cream-colored and cottony mycelium. Conidiophores usually arising from aerial hyphae, with a single phialide or whorls of two to four phialides, 7.0–29.1 × 1.8–2.6, mean = 12.5 × 2.4 μm. Phialides 7.9–26.3 × 1.8–3.4 μm, mean = 12.7 × 2.6 μm, smooth-walled, swollen at the base or slightly flask-shaped, and then tapering to a thin neck. Conidial arrangement *Evlachovaea*-like. Conidia smooth and hyaline, oblong-elliptical, 3.9–6.1 × 2.0–2.5 μm, mean ± SD =4.5 ± 0.4 × 2.3 ± 0.1 μm, L/W ratio = 2.0, n = 30. Sexual morph not observed.

**Figure 6 jof-11-00028-f006:**
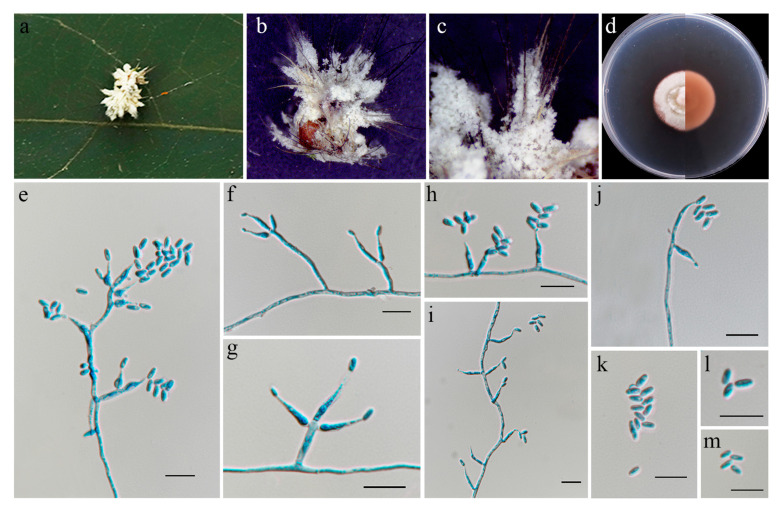
*Blackwellomyces lateris* (HMAS 352953). (**a**) Larvae of Lepidoptera infected by *B. lateris*. (**b**,**c**) Close-up of specimen. (**d**) Upper and reverse view of colony after incubation for 14 days on PDA. (**e**–**j**) Conidiogenous cells (conidiophores and phialides). (**k**–**m**) Conidia. Scale bars: (**e**–**m**) 10 µm.

Culture characteristics: Colonies on PDA moderately fast-growing, 24 mm diameter in 14 days at 25 °C, growth rate 1.7 mm/day, white mycelium, cottony with low mycelium density, and producing yellowish–brown pigment diffusing in the medium.

Material examined: China, Fujian Province, Sanming City, Mingxi County, Xiayang Township, 117°27′33″ E, 26°20′46″ N, on the larvae of Lepidoptera buried in soil, 30 August 2023, paratype HMAS 352953;ex-paratype living culture CGMCC3.27468.

Notes: *Blackwellomyces lateris* was originally described in Lepidoptera larvae in Thailand, Chiang Mai Province [47]. The strain identified as *B. lateris* (CGMCC3.27468) in the present study was also isolated from Lepidoptera larvae. Phylogenetic analysis showed that strain CGMCC3.27468 formed a branch sister to *B. lateris* (MFLU 18-0663/MRC170617) with relatively strong statistical support (100% ML/1 PP, Figure 1). CGMCC3.25363 was similar to the latter in SSU (with 99.90% sequence identity), LSU (99.87%), *tef1-α* (100%), *rpb1* (99.70%), and *rpb2* (99.91%) sequences. Therefore, we identified this fungus as *B. lateris*, marking the first record of *B. lateris* in China.

#### 3.4.4. *Liangia sinensis*, Yu, H.; Wang, Y.B.; Wang, Y.; Fan, Q.; Yang, Z.L. Fungal Diversity **2020**, *103*, 1–46 [2]; Figure 7

Description: *Lecanicillium*-like. Strains isolated from the stromata (i.e., hyper-parasitic) of *Ophiocordyceps globiceps* (Ophiocordycipitaceae, Hypocreales) infecting a fly (Diptera). Hyphae hyaline, septate, branched, smooth-walled, and 1.4–1.9 µm wide. Phialides lanceolate, occurring directly from the prostrate hyphae, solitary, and gradually attenuated toward the apex, 13.8–30.6 × 1.1–1.8 μm, mean = 21.5 × 1.4 μm. Two conidia types, hyaline, smooth-walled, with macro- and microconidia, aseptate, both existing alone or in pairs at the apex of phialides. Macroconidia oblong–oval to fusiform, 7.2–11.1 × 1.2–1.7 µm, mean ± SD = 8.7 ± 1.0 ×1.5 ± 0.1 μm, L/W ratio = 5.8, n = 30. Microconidia oval to oblong, 3.2–5.5 × 1.1–1.9 μm, mean ± SD = 4.4 ± 0.5 ×1.4 ± 0.2 μm, L/W ratio = 3.1, n = 30. Sexual morph not observed.

**Figure 7 jof-11-00028-f007:**
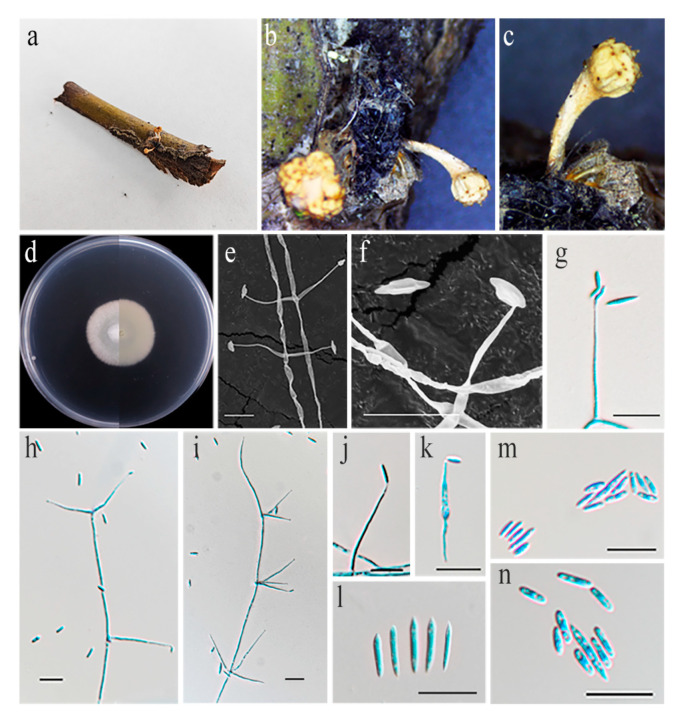
*Liangia sinensis* (HMAS 352686). (**a**–**c**) Stromata of *Ophiocordyceps globiceps* arising from fly (Muscidae, Diptera) clinging to decaying wood, from which *L. muscidarum* was isolated. (**d**) Upper and reverse view of colony after incubation for 14 days on PDA. (**e**–**k**) Conidiogenous cells (conidiophores and phialides). (**l**–**n**) Conidia. Scale bars: (**e**–**n**) 10 µm.

Culture characteristics: Colonies on PDA slow-growing, 21–25 mm in diameter after 14 days at 25 °C, growth rate 1.5–1.8 mm/day, white and cottony, consisting of a basal felt, with slightly raised mycelial density at the centre.

Material examined: China, Jilin Province, Baishan City, 127°59′44.3″ E, 42°33′48.8″ N, on *Ophiocordyceps globiceps* on fly (Muscidae, Diptera) clinging to decaying wood, 2 September 2023, paratype HMAS 352686; ex-paratype living culture CGMCC3.25667.Ibid., paratype HMAS 298129; ex-paratype living culture CGMCC3.27196.

Notes: Wang et al. reported *Liangia sinensis* isolated from the entomopathogenic fungus *Beauveria yunnanensis* as the type species of the genus *Liangia* [2]. At present, the sexual morph of this genus has not been determined. Phylogenetic analysis of five genes showed that two strains (CGMCC3.25667 and CGMCC3.27196) tightly cluster with *L. sinensis* in the combined phylogenetic tree (Figure 1). Strain characteristics were very closely linked to *L. sinensis*, which also shows two types of conidia (macroconidia: 7.2–11.1 × 1.2–1.7 µm vs. 4.5–9.3 × 1.2–1.9 µm; microconidia: 3.2–5.5 × 1.1–1.9 μm vs. 1.8–3.3 × 1.1–1.8 µm) and a phialide (13.8–30.6 × 1.1–1.8 μm vs. 16.7–59.0 × 0.7–1.6 μm). Additionally, CGMCC3.25667 and CGMCC3.27196 were highly similar to *L. sinensis* in SSU (with 100% sequence identity), LSU (99.63%), *tef–1α* (99.89%), *rpb1* (99.56%), and *rpb2* (99.79%) sequences. Notably, isolates CGMCC3.27196 and CGMCC3.27197 were found parasitizing the stromata of *Beauveria yunnanensis* associated with Lepidoptera pupa, whereas *Liangia sinensis* (YFCC 3103 and YFCC 3104) was found on an *Ophiocordyceps globiceps*-infected dipteran host. The results of our study show that the *L. sinensis* can be isolated from different hosts.

#### 3.4.5. *Niveomyces multisynnematus* Tasanathai, Noisripoom and Kobmoo, Fungal Systematics and Evolution **2023**, *12*, 91–110 [7]; Figure 8

Description: White and cottony mycelium forming on the stromata (i.e., hyper-parasitic) of *Ophiocordyceps* sp. (Ophiocordycipitaceae, Hypocreales), with multiple protuberances on the surface of the apex of the stromata. Synnemata simple, indeterminate, cylindrical, curved, 2–3 mm long, and 25–60 µm wide. Hyphae septate, hyaline, smooth-walled, irregularly branched, and 1.8–2.6 µm wide. Conidiogenous cells arising directly from the hyphae cylindrical, sometimes capitate, and bearing an irregularly geniculate rachis, 8.3–44.3 × 1.0–2.2 µm, mean = 27.4 × 1.5 μm. Conidia hyaline, aseptate, smooth-walled, oblong to fusiform with rounded ends and an apiculus, exists on the denticles of conidiogenous cells, 3.0–5.3 × 1.4–2.2 μm, mean ± SD = 4.6 ± 0.5 ×2.0 ± 0.2 μm, L/W ratio = 2.3, n = 50. Sexual morph not observed.

**Figure 8 jof-11-00028-f008:**
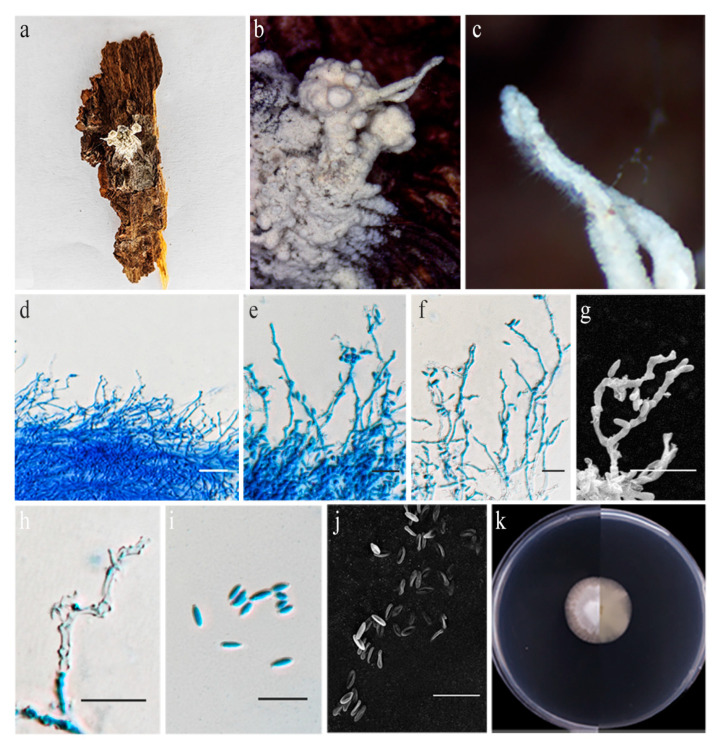
*Niveomyces multisynnematus* (HMAS 352684). (**a**) *N. multisynnematus* growing on *Ophiocordyceps* sp. (**b**,**c**) Close-up of synnemata. (**d**–**h**) Conidiogenous cells. (**i**,**j**) Conidia. (**k**) Upper and reverse view of colony after incubation for 14 days on PDA.Scale bars: (**d**–**j**) 10 µm.

Culture characteristics: Colonies on PDA slow-growing, 18–23 mm in diameter after 14 days at 25 °C, growth rate 1.3–1.6 mm/day, white and cottony, medium density, slightly raised, consisting of a basal felt, and the reverse colony gradually shifted from white to light yellow over time.

Material examined: China, Jilin Province, Baishan City, 127°59′59.3″ E, 42°33′58.1″ N, on *Ophiocordyceps* sp. on fly (Muscidae, Diptera)clinging to decaying wood, 2 September 2023, paratype HMAS 352685; ex-paratype living culture CGMCC3.25670.Ibid., paratype HMAS 352684; ex-paratype living culture CGMCC3.25672.

Notes: *Niveomyces multisynnematus* was originally found on leafhoppers (Hemiptera) on the underside of leaves in Thailand. Our multi-locus phylogenetic analyses indicate that strains CGMCC3.25670 and CGMCC3.25672fall within *Niveomyces*, clustering tightly with *N.multisynnematus* with strong support (100% ML/1 PP, Figure 1). Notably, the strains characterized herein form multiple synnemata on the stromata of *Ophiocordyceps* sp. (parasitizing a fly), whereas *N. multisynnematus* produces synnemata at various locations on *Ophiocordyceps aff. flavida*. Morphologically, our strains were highly similar to *N.multisynnematus*, including the conidia and conidiogenous cells (3.0–5.3 × 1.4–2.2 µm vs. 2.0–5.0 × 1.0–1.5 µm; 8.3–44.3 × 1.0–2.2 µm vs. 20–50 × 1 µm). Additionally, the two isolates characterized herein were highly similar to *N.multisynnematus* in SSU (with 99.89% sequence identity), LSU (100%), *tef–1α* (99.31%), *rpb1* (100%), and *rpb2* (100%) sequences. Therefore, we identify our isolates as *N. multisynnematus*, marking the first record of *N. multisynnematus* in China.

## 4. Discussion

### 4.1. Taxonomy and Phylogeny of Cordycipitaceae

Species within the Cordycipitaceae are widely distributed and include a stunning variety of entomopathogenic fungi that have significant ecological impacts as well as economic value, which has led to increased attention towards their taxonomic placement and identification. With the study of entomopathogenic fungi entering the genomic era, molecular methods have been applied to the systematics and evolutionary biology of Cordycipitaceae, and the classification of this family has undergone a major shift from polyphyletic to monophyletic [1,3,22].

Our data indicate that the fungal specimens collected belong to Cordycipitaceae, and group within the *Liangia*, *Niveomyces*, and *Samsoniella* genera. In total, two new species (*Akanthomyces baishanensis* sp. nov. and *Samsoniella sanmingense* sp. nov.), two new record species (*Blackwellomyces lateris* and *Niveomyces multisynnematus*) and one known species (*Liangia sinensis*) are described.

*Samsoniella* was assigned by Mongkolsamrit et al. in 2018 for the species *S. inthanonensis* (the type species) and two other species, *S. alboaurantia* and *S. aurantia* [17]. *Samsoniella* has a wide geographical distribution, with 38 names currently recorded in the Index Fungorum (http://www.indexfungorum.org/; accessed 1 September 2024). In China, the majority of *Samsoniella* species are found in warm and humid climates, such as the Yunnan and Guizhou provinces [2,48]. In Yunnan, based on multiple gene sequences (SSU, LSU, *tef-1α*, *rpb1*, and *rpb2*), Wang et al. found *Samsoniella* species to be important for economic, medical, and ecological values, i.e., *S. alpina*, *S. antleroides*, *S. cardinalis*, *S. cristata*, *S. lanmaoa*, *S. kunmingensis*, *S. ramosa*, *S. tortricidae*, and *S. yunnanensis* [2]. In Guizhou, Chen et al. [20] isolated *S. hymenopterorum*, *S. coleopterorum*, and *S. lepidopterorum* from Hymenoptera larvae, Coleopteran larvae, and Lepidoptera pupae, respectively [48]. *S. pupicola* was then identified in several *Isaria farinosa* strains [20]. Recently, it has been reported that some *Samsoniella* members (*S. formicae*, *S. erucae*, and *S. neoupicola*) appear in unique karst ecological environments. Subsequently, Chen et al. discovered *S. duyunensis* and *S. vallis*, which provided new insights into the diversity and ecological adaptability of *Samsoniella* species in different habitats [49]. Here, we report a new species, *S. sanmingense* sp. nov., isolated from the pupae of Lepidoptera. *S. sanmingense* can be distinguished from *S. tiankengensis* by a combination of broader phialides and bigger oblong to fusiform conidia, and can also be distinguished from other known species of the genus by the morphology of their colonies, as well as by phylogenetic, morphological, and ecological features. It is worth mentioning that we also isolated *B. lateris* from Lepidoptera larvae for the first time in China, which is the first time that an asexual morph of this genus has been found. The tropical coastal climate of Fujian province, located on the southeastern coast of China, proves to have a rich diversity that merits further exploration of entomopathogenic fungi [50,51].

*Akanthomyces*, *Samsoniella*, and other genera with *Isaria*-like or *Lecanicillium*-like morphs have similar morphs characteristics, so it is often difficult to distinguish these species using only morphological characteristics [17]. Due to the difficulty of distinguishing species based solely on morphological characteristics, Chen et al. [49] suggested that modern identification of the *Akanthomyces* genus can utilize any two of three proscribed loci (e.g., ITS, LSU, and/or *rpb2*) in phylogenetic analysis to differentiate and interpret these closely related species. Subsequently, Khonsanitet al. [4] conducted a comprehensive phylogenetic and morphological examination of the species within *Akanthomyces*, and reclassified these species into four genera based on the type of substrate or the taxonomic group of the host. Currently, 13 species are classified within *Akanthomyces*, all of which are exclusively associated with moth infections. Here, we have used a combined SSU, LSU, *tef1-α*, *rpb1*, and *rpb2* sequence dataset to support the identification of a new species that has been named *Ak. baishanensis*. *Ak. baishanensis* was found on adult moths in Northeast China. *Ak. baishanensis* forms an independent clade distinct from *Ak. phariformis*, *Ak. bannaensis*, and *Ak. taiwanicus*, with strong support value. Khonsanit et al. [4] described *Ak. phariformis* based on specimens collected in Tak Province, Thailand, describing the sexual morph characteristics of this species, including ovoid perithecia and filiform ascospores. Compared to *Ak. phariformis*, *Ak. baishanensis* has larger perithecia. When compared to *Ak. taiwanicus*, *Ak. baishanensis* exhibits longer phialides and wider conidia. According to the morphological description of Zhanget al. [52], *Ak. bannaensis* was found in adults of *Dudusa sphingiformis* on the underside of the leaves of angiosperms, with wider phialides and larger ellipsoidal or obovoid conidia. In contrast, the presence of this family has not been reported in Northeast China, and its occurrence in this area may be related to the impact of global warming on the distribution patterns of fungi.

Among these species, two are particularly striking and eye-catching. During the exploration of entomopathogenic fungi in Jilin Province, we discovered a new specimen with *Sporothrix*-like asexual morphs, and this genus has been reported for the first time in China. *Niveomyces* belongs to a relatively specialized lineage within Cordycipitaceae that has evolved as specialized mycoparasites. Scattered and denticulate conidiogenous cells are typical of the *Niveomyces* species, andare observed in all members of this genus as well as in the species we have described [7,23]. The genus *Liangia* was assigned to Cordycipitaceae with *Liangia sinensis* as the type species, and currently the genus includes only a single species. Wang et al. only observed the asexual morph of *Liangia*. In this study, we isolated a known species *Liangia sinensis* from *Ophiocordyceps globiceps* parasitizing a Diptera host [2]. Interestingly, we found that *L. sinensis* and *N. multisynnematus* derived from dead insects infected with *Ophiocordyceps* sp., indicating a strong parasitic relationship. Meanwhile, our results further support the hypothesis that *Liangia sinensis* may be a hyper-parasitic fungus of *Beauveria yunnanensis* [2]. Jilin Province, located in Northeast China, has a temperate continental climate and abundant resources [53]. However, thus far, the diversity of Cordycipitaceae in this region has not yet been reported. Biogeographically, most species within the *Akanthomyces*, *Liangia*, and *Niveomyces* are primarily distributed in tropical and subtropical regions, whereas the three species reported here have mainly been collected in temperate continental climates. These reports, along with our own findings, may indicate that Cordycipitaceae species are more widely distributed than previously recognized. This provides new insights into the diversity and ecological adaptability of these species in different habitats.

### 4.2. Possible Origin of Akanthomyces Sensu Lato

In recent years, molecular dating methods have been widely applied, leading to significant advancements in estimating fungal divergence times, which can enhance our understanding of evolutionary processes across all taxonomic levels. Based on the concatenated sequence dataset of ITS, LSU, *rpb2*, *tef1–α*, and *tub2*, it has been proposed that the common ancestor of Pseudoplagiostomataceae and Apoharknessiaceae appeared in the Cretaceous period (104.1 Mya, 86.0–129.0 Mya) [54]. Similarly, the common ancestor of the eight branches of *Lecanicillium* appears to have occurred ~84.36 Mya [38], with the proposed mean stem age of *Favolaschia* as 49.6 Mya (95% HPD of 36.7–75.7 Mya) [55]. Recently, Mu et al. [56] proposed that the ancestor of Schizoparmaceae appeared during the Upper Cretaceous period, approximately 75.7 Mya (95% HPD: 60.3–91.3 Mya).

Molecular diversity analyses of macrofungi, animals, and plants have been widely applied for biogeographical analyses aimed at identifying likely ancestral origin points. However, the study of the origin and historical distribution of many (filamentous) fungal species has received less attention. *Akanthomyces s.l.* is an important entomopathogenic fungus with a worldwide distribution and diverse bioactive substances, and here, we used RASP in phylogeny and selected the Statistical Dispersal–Extinction–Cladogenesis (S–DEC) model to reconstruct the historical biogeography of this genus. Using two ribosomal RNA genes (SSU and LSU) and three protein-coding genes (*tef1–α*, *rpb1*, and *rpb2*), we inferred that *Akanthomyces s.l.* likely originated in Asia between the Oligocene and Eocene periods, with an average stem age of 34.57 Mya (95% highest posterior density of 31.41–37.67 Mya). Asia is inferred to be the most probable ancestral area, with dispersals into Africa, Europe, North America, and South America. We found that the diversity of *Akanthomyces s.l.* was not directly related to the dramatic extinction event of the Cretaceous–Tertiary period, similar to previous literature reports [57]. This may be due to long-term environmental adaptation and co-evolution with insects, or random changes in temperature and aridification in the Miocene might have caused populations of many organisms to expand and contract [38,58].

Asia is the apparent center of origin for the “Group I” set (13.46 Mya, 95% HPD 11.67–15.31 Mya), which consists of seven species: *Arachnidicolaaraneicola*, *Ar. araneogenum*, *Ar. bashanensis*, *Ar. beibeiensis*, *Ar. kunmingensis*, *Ar. subaraneicola*, and *Ar. tiankengensis*, from China, four species: *Ar. kanyawimiae*, *Ar. sulphureus*, *Ar. thailandica*, and *Ar. waltergamsii* from Thailand, and one species: *Ar. kanyawimiae*, from Japan. The population of the ancestral species in Group I shows a trend of migrating towards small regions (refugia) in East Asia (mainly in central and southwest China), which may be because the Himalayan mountains and Qinghai–Tibet Plateau experienced a series of important geographical events during the Neogene period, resulting in a significant increase in their altitude [59,60].

Asia and South America appear to be the center of origin for the “Group II” set (17.29 Mya, 95% HPD 15.13–19.48 Mya), with dispersal to Europe and North America, resulting in high endemism of *Akanthomyces s.l.* in the region. Four species, *Lecanicillium neocoleopterorum*, *L. attenuatum*, *L. pissodis*, and *L. muscarium*, are closely related, suggesting the dispersal route of Asia–Europe–North America. *L. lecanii* and *L. sabanense* show significant differences from other taxa in this group, which suggests an Asia–South America dispersal route. One hypothetical mechanism for this dispersal from Asia to South America may be via hurricanes and typhoons that circulate the South Pacific Ocean. The transoceanic or long-distance dispersal (LDD) hypothesis has been proposed to explain biogeographical patterns and the distribution of animals, plants, and fungi, but this mechanism is considered rare [61].

Bayesian binary analysis supported Asia as the center of origin for the “Group III” set (17.91 Mya, 95% HPD 16.18–19.81 Mya), with nine *Akanthomyces* species found in Southeast Asia, and four species found in China. *Ak. xixiuensis*, *Ak. baishanensis*, *Ak. Bannaensis*, and *Ak. taiwanicus* are from Guizhou, Jilin, Yunnan, and Taiwan provinces in China, respectively, while *Ak. buriramensis*, *Ak. fusiformis*, *Ak. laosensis*, *Ak. niveus*, *Ak. noctuidarum*, *Ak. phariformis*, *Ak. Pseudonoctuidarum*, and *Ak. pyralidarum* are mainly restricted to tropical regions in Southeast Asia. *Ak. aculeatus* exhibits a continuous distribution in Europe and Ecuador; therefore, we deduced that the occurrence of *A. aculeatus* in South America may be due to recent human-mediated introductions. In addition, two species, *Ak. tuberculatus* and *Ak. aculeatus*, show significant differences in geographical distribution from other taxa in this group. We propose that the increase in tropical forest area associated with the Miocene warming trend may have played a crucial role in facilitating species exchanges between Europe and Asia [56]. These observations strongly support early diversification of *Akanthomyces s.l.* species in Asia. The characterization of additional species, coupled with fossil evidence, is needed to further support this hypothesis, and additional research on the biogeography of the Cordycipitaceae is warranted. Overall, our data provide a significant addition to the diversity, ecology, and evolutionary aspects of the Cordycipitaceae, and suggest that significant additional discovery awaits.

## 5. Conclusions

In this study, based on the combination of morphological observations and multi-locus phylogenetic analyses, we identified two new species, two new record species, and one known species of Cordycipitaceae distributed within five different genera from China, viz. *Akanthomyces baishanensis* sp. nov., *Samsoniella sanmingense* sp. nov., *Blackwellomyces lateris*, *Niveomyces multisynnematus*, and *Liangia sinensis*. Molecular clock analyses examining divergence times, along with historical biogeographic analyses, suggest that the ancestor of *Akanthomyces sensu lato* (*s.l.*) most likely originated in Asia during the Paleogene period, approximately 34.57 Mya. These findings provide valuable help for the classification, diversity, and evolutionary aspects of Cordycipitaceae species.

## Figures and Tables

**Figure 1 jof-11-00028-f001:**
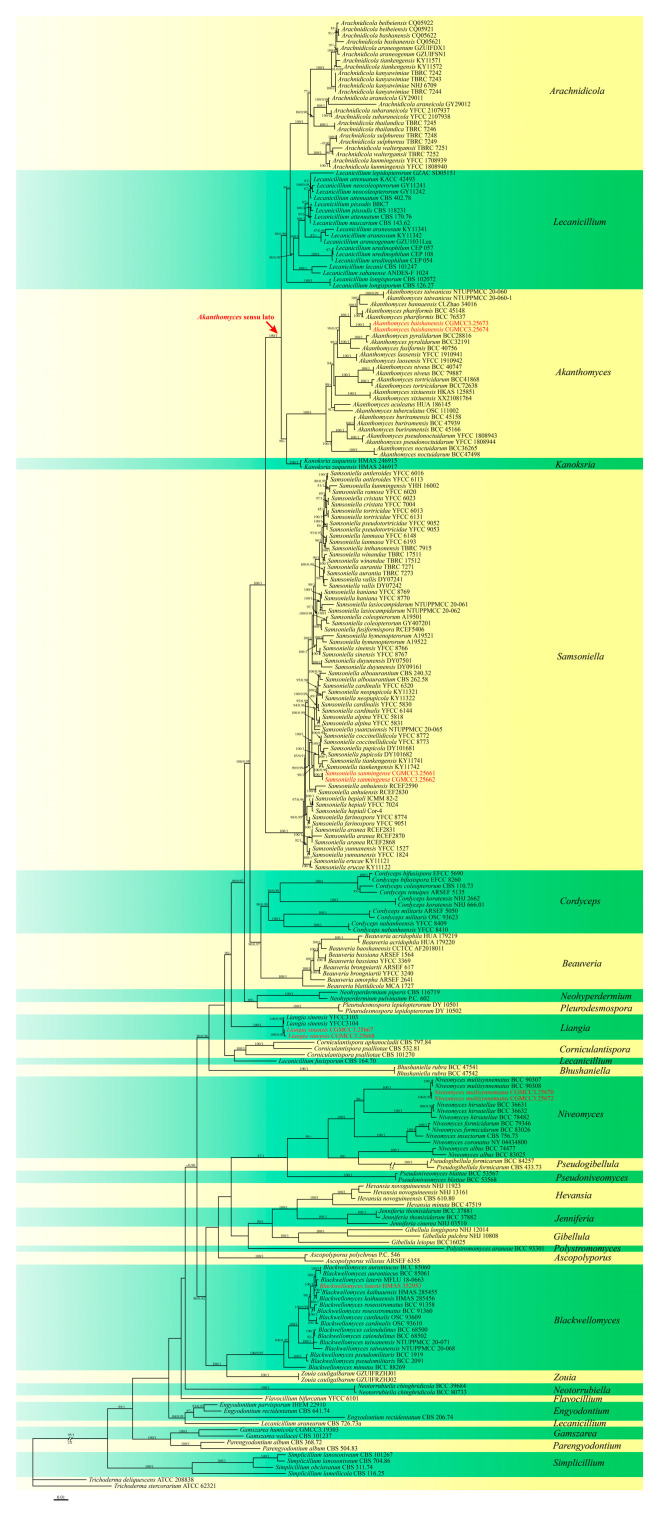
Phylogeny of Cordycipitaceae inferred from the combined dataset SSU, LSU, *tef–1α*, *rpb1*, and *rpb2*, with *Trichoderma deliquescens* ATCC 208838 and *Trichoderma stercorarium* ATCC 62321 as outgroup. The maximum likelihood bootstrap support values and Bayesian posterior probabilities bootstrap support values above 70% and 0.90 are shown at the first and second position, respectively. Fungal isolates from this study are shown in red. Arrows show the support values at the branching points. The scale at the bottom left indicates the number of nucleotide substitutions per site. Yellow–green strips represent different neighboring species.

**Figure 2 jof-11-00028-f002:**
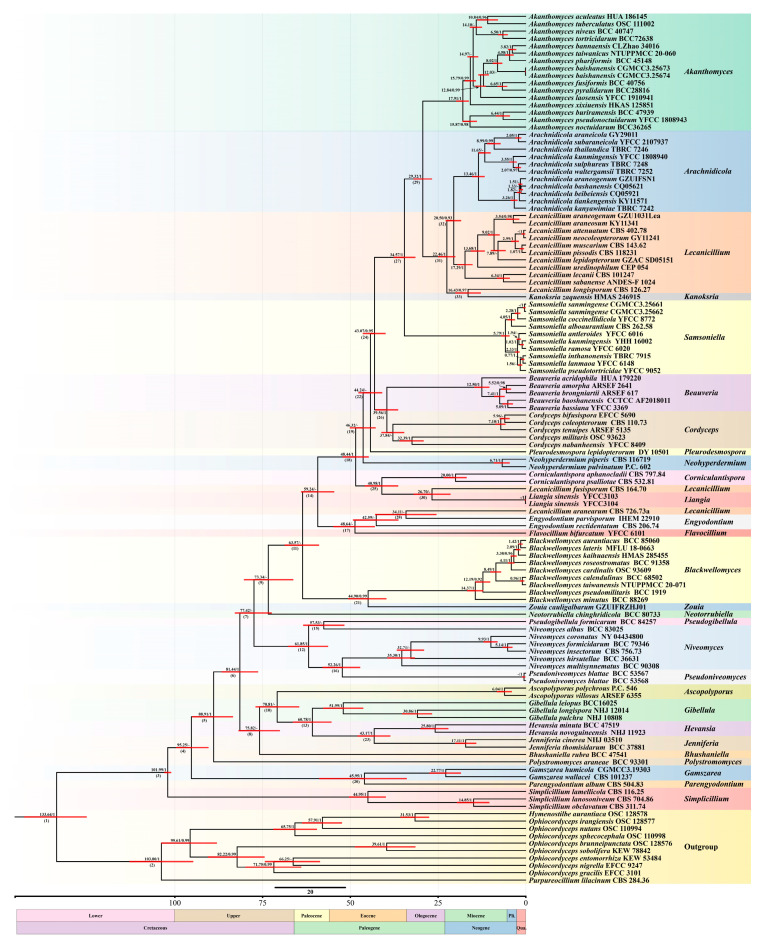
Divergence time estimation of Cordycipitaceae inferred from molecular clock analyses based on five-locus dataset of SSU, LSU, *tef–1α*, *rpb1*, and *rpb2* regions. The 95% highest posterior density (HPD) and Bayesian posterior probabilities (BYPP) of estimated divergence times indicated by horizontal red bars. The mean divergence times of each node not less than 0.7 are shown at the internodes. Scale in millions of years (Mya).

**Figure 3 jof-11-00028-f003:**
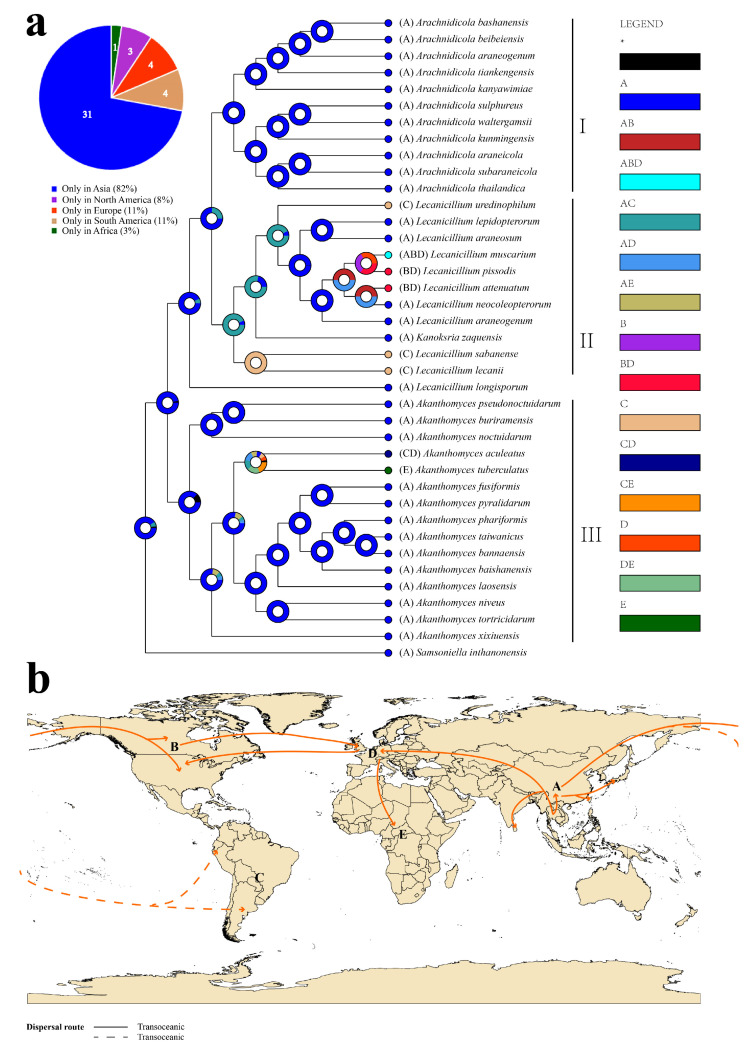
Ancestralstate reconstruction and divergence time estimation of *Akanthomyces s.l.* using the SSU, LSU, *tef–1α*, *rpb1*, and *rpb2* dataset. (**a**) The pie chart in each node indicates the possible ancestral distributions inferred from Bayesian Binary Markov chain Monte Carlo analysis (BBM) implemented in RASP. A black asterisk represents other ancestral ranges. (**b**) Possible dispersal routes of *Akanthomyces s.l.*

**Table 1 jof-11-00028-t001:** PCR primers and their annealing temperatures used in this study.

Gene	Primers	Sequence (5′-3′)	Annealing Temperature	References
SSU	nrSSU-CoF	TCTCAAAGATTAAGCCATGC	55 °C	[24]
	nrSSU-CoR	TCACCAACGGAGACCTTG		
LSU	LR0R	GTACCCGCTGAACTTAAGC	52 °C	[25,26]
	LR5	TCCTGAGGGAAACTTCG		
*tef-1α*	EF1α-EF	GCTCCYGGHCAYCGTGAYTTYAT	60 °C	[27]
	EF1α-ER	ATGACACCRACRGCRACRGTYTG		
*rpb1*	RPB1-Ac	CAYCCWGGYTTYATCAAGAA	From 57 °C to 72 °C at 0.2 °C/s	[28,29]
	RPB1-Cr	CCNGCDATNTCRTTRTCCATRTA		
*rpb*2	fRPB2-5F	GAYGAYMGWGATCAYTTYGG	56 °C	[30]
	fRPB2-7cR	CCCATRGCTTGTYYRCCCAT		

**Table 2 jof-11-00028-t002:** Taxa used in the phylogenetic analysis and their GenBank accession numbers.

Species	Strain No.	Host/Substrate	GenBank Accession Numbers				
			SSU	LSU	*tef-1α*	** *rpb1* **	** *rpb2* **
*Akanthomyces aculeatus*	HUA 186145	Lepidoptera	MF416572	MF416520	MF416465	-	-
** *Akanthomyces baishanensis* **	**CGMCC3.25673**	**Lepidoptera; Noctuidae**	**PP177381**	**PP179404**	**PP464678**	**PP464641**	**PP464655**
** *Akanthomyces baishanensis* **	**CGMCC3.25674**	**Lepidoptera; Noctuidae**	**PP177382**	**PP179405**	**PP464679**	**PP464642**	**PP464656**
*Akanthomyces bannaensis*	CL Zhao 34016	Lepidoptera; *Dudusa sphingiformis*	-	PP571897	-	-	PP588774
*Akanthomyces buriramensis*	BCC 45158	Lepidoptera; Noctuidae	-	ON008543	ON013546	ON013561	-
*Akanthomyces buriramensis*	BCC 45166	Lepidoptera; Noctuidae	-	ON008544	ON013547	ON013562	-
*Akanthomyces buriramensis*	BCC 47939	Lepidoptera; Noctuidae	-	ON008545	ON013548	ON013563	-
*Akanthomyces fusiformis*	BCC 40756	Lepidoptera; Pyralidae	-	ON008549	ON013552	ON013567	ON013576
*Akanthomyces laosensis*	YFCC 1910941	Lepidoptera; Noctuidae	-	OQ509510	OQ506286	OQ511535	OQ511549
*Akanthomyces laosensis*	YFCC 1910942	Lepidoptera; Noctuidae	-	OQ509511	OQ506287	OQ511536	OQ511550
*Akanthomyces niveus*	BCC 40747	Lepidoptera	-	ON008550	ON013553	ON013568	ON013577
*Akanthomyces niveus*	BCC 79887	Lepidoptera	-	ON008551	ON013554	-	ON013578
*Akanthomyces noctuidarum*	BCC36265	Lepidoptera; Noctuidae	-	MT356084	MT477978	MT477994	MT477987
*Akanthomyces noctuidarum*	BCC47498	Lepidoptera; Noctuidae	-	MT356086	MT477980	MT477996	MT477988
*Akanthomyces phariformis*	BCC 45148	Lepidoptera; Erebidae	-	ON008556	ON013559	-	ON013583
*Akanthomyces phariformis*	BCC 76537	Lepidoptera; Noctuidae	-	ON008557	ON013560	-	ON013584
*Akanthomyces pseudonoctuidarum*	YFCC 1808943	Lepidoptera; Noctuidae		OQ509512	OQ506288	OQ511537	OQ511551
*Akanthomyces pseudonoctuidarum*	YFCC 1808944	Lepidoptera; Noctuidae		OQ509513	OQ506289	OQ511538	OQ511552
*Akanthomyces pyralidarum*	BCC28816	Lepidoptera; Pyralidac	-	MT356091	MT477982	MT478000	MT478007
*Akanthomyces pyralidarum*	BCC32191	Lepidoptera; Pyralidac	-	MT356092	MT477983	MT478001	MT477989
*Akanthomyces taiwanicus*	NTUPPMCC 20-060	Lepidoptera (*Agrius convolvuli*)	-	MT974356	MW200213	MW200221	MW200230
*Akanthomyces taiwanicus*	NTUPPMCC 20-060-1	Lepidoptera (*Agrius convolvuli*)	-	PP190926	-	-	-
*Akanthomyces tortricidarum*	BCC41868	Lepidoptera; Tortricidae	-	MT356089	MT477985	MT477998	MT478008
*Akanthomyces tortricidarum*	BCC72638	Lepidoptera; Tortricidae	-	MT356088	MT478004	MT477997	MT477992
*Akanthomyces tuberculatus*	OSC 111002	Lepidoptera	DQ522553	DQ518767	DQ522338	-	DQ522435
*Akanthomyces xixiuensis*	HKAS 125851	Lepidoptera	OP693479	OP693481	OP838888	OP838890	OP838892
*Akanthomyces xixiuensis*	XX21081764	Lepidoptera	OP693478	OP693480	OP838887	OP838889	OP838891
*Arachnidicola araneicola*	GY29011	Araneae; spider	-	-	MK955950	MK955944	MK955947
*Arachnidicola araneicola*	GY29012	Araneae; spider	-	-	MK955951	MK955945	MK955948
*Arachnidicola araneogenus*	GZUIFDX1	*Araneus* sp.	-	-	-	MH978181	MH978184
*Arachnidicola araneogenus*	GZUIFSN1	*Araneus* sp.	-	-	MH978188	MH978183	MH978186
*Arachnidicola bashanensis*	CQ05621	Araneae; spider	-	OQ300420	OQ325024	-	OQ349684
*Arachnidicola bashanensis*	CQ05622	Araneae; spider	-	OQ300421	OQ325025	-	OQ349685
*Arachnidicola beibeiensis*	CQ05921	Araneae; spider	-	OQ300424	OQ325028	-	OQ349688
*Arachnidicola beibeiensis*	CQ05922	Araneae; spider	-	OQ300427	OQ325029	-	OQ349689
*Arachnidicola kanyawimiae*	NHJ 6709	Araneae; spider	EU369110	EU369042	EU369025	EU369067	EU369086
*Arachnidicola kanyawimiae*	TBRC 7242	Araneae; spider	-	MF140718	MF140838	MF140784	MF140808
*Arachnidicola kanyawimiae*	TBRC 7243	On unidentified insect	-	MF140717	MF140837	MF140783	MF140807
*Arachnidicola kanyawimiae*	TBRC 7244	Araneae; spider	-	MF140716	MF140836	-	-
*Arachnidicola kunmingensis*	YFCC 1708939	Araneae; spider	-	OQ509508	OQ506284	OQ511533	OQ511547
*Arachnidicola kunmingensis*	YFCC 1808940	Araneae; spider	-	OQ509509	OQ506285	OQ511534	OQ511548
*Arachnidicola subaraneicola*	YFCC 2107937	Araneae; spider	-	OQ509514	OQ506290	OQ511539	OQ511553
*Arachnidicola subaraneicola*	YFCC 2107938	Araneae; spider	-	OQ509515	OQ506291	OQ511540	OQ511554
*Arachnidicola sulphureus*	TBRC 7248	Araneae; spider	-	MF140722	MF140843	MF140787	MF140812
*Arachnidicola sulphureus*	TBRC 7249	Araneae; spider	-	MF140721	MF140842	MF140786	MF140734
*Arachnidicola thailandica*	TBRC 7245	Araneae; spider	-	-	MF140839	-	MF140809
*Arachnidicola thailandica*	TBRC 7246	Araneae; spider	-	MF140719	MF140840	-	MF140810
*Arachnidicola tiankengensis*	KY11571	Araneae; spider	-	ON502825	ON525447	-	ON525446
*Arachnidicola tiankengensis*	KY11572	Araneae; spider	-	ON502827	ON525449	-	ON525448
*Arachnidicola waltergamsii*	TBRC 7251	Araneae; spider	-	MF140713	MF140833	MF140781	MF140805
*Arachnidicola waltergamsii*	TBRC 7252	Araneae; spider	-	MF140714	MF140834	MF140782	MF140806
*Ascopolyporus polychrous*	P.C. 546	Hemiptera	-	DQ118737	DQ118745	DQ127236	-
*Ascopolyporus villosus*	ARSEF 6355	Hemiptera	-	AY886544	DQ118750	DQ127241	-
*Beauveria acridophila*	HUA 179219	Orthoptera; Acrididae	-	JQ895541	JQ958613	JX003857	JX003841
*Beauveria acridophila*	HUA 179220	Orthoptera; Acrididae	JQ895527	JQ895536	JQ958614	JX003852	JX003842
*Beauveria amorpha*	ARSEF 2641	Hymenoptera; Formicidae	-	AB100039	AY531917	HQ880880	HQ880952
*Beauveria baoshanensis*	CCTCC AF2018011	Coleoptera; Chrysomelidae	MG642882	MG642840	MG642897	MG642854	MG642867
*Beauveria bassiana*	ARSEF 1564	Lepidoptera; Arctiidae	EU334676	-	HQ880974	HQ880833	HQ880905
*Beauveria bassiana*	YFCC 3369	Coleoptera; Scarabaeidae	MN576768	MN576824	MN576994	MN576884	MN576938
*Beauveria blattidicola*	MCA 1727	Blattodea; Blattidae	MF416593	MF416539	MF416483	MF416640	-
*Beauveria brongniartii*	ARSEF 617	Coleoptera; Scarabaeidae	AB027335	AB027381	HQ880991	HQ880854	HQ880926
*Beauveria brongniartii*	YFCC 3240	Coleoptera; Scarabaeidae	MN576769	MN576825	MN576995	MN576885	MN576939
*Bhushaniella rubra*	BCC 47541	Spider eggs in sac	-	OQ892133	OQ914428	OQ914431	OQ914433
*Bhushaniella rubra*	BCC 47542	Spider eggs in sac	-	OQ892134	OQ914429	OQ914432	OQ914434
*Blackwellomyces aurantiacus*	BCC 85060	Lepidoptera (larva)	-	MT003028	MT017840	MT017800	MT017819
*Blackwellomyces aurantiacus*	BCC 85061	Lepidoptera (larva)	-	MT003029	MT017841	MT017801	MT017820
*Blackwellomyces calendulinus*	BCC 68500	Coleoptera (larva)	-	MT003030	MT017842	MT017802	MT017821
*Blackwellomyces calendulinus*	BCC 68502	Coleoptera (larva)	-	MT003031	MT017843	MT017803	MT017822
*Blackwellomyces cardinalis*	OSC 93609	Lepidoptera; Tineidae	AY184973	AY184962	DQ522325	DQ522370	DQ522422
*Blackwellomyces cardinalis*	OSC 93610	Lepidoptera; Tineidae	AY184974	AY184963	EF469059	EF469088	EF469106
*Blackwellomyces kaihuaensis*	HMAS 285455	Lepidoptera (larva)	OQ981975	OQ981968	OQ980401	OQ980409	OQ980408
*Blackwellomyces kaihuaensis*	HMAS 285456	Lepidoptera (larva)	OQ981976	OQ981969	OQ980402	OQ980410	-
*Blackwellomyces lateris*	MFLU 18-0663	Coleoptera (larva)	NG_067678	MK086061	MK069471	MK084615	MK079354
** *Blackwellomyces lateris* **	**HMAS 352953**	**Lepidoptera (larva)**	**PP554483**	**PP554184**	**PP556330**	**PP556331**	**PP556332**
*Blackwellomyces minutus*	BCC 88269	Coleoptera (larva)	-	MT003032	MT017844	MT017804	MT017823
*Blackwellomyces pseudomilitaris*	BCC 1919	Lepidoptera (larva)	MF416588	MF416534	MF416478	-	MF416440
*Blackwellomyces pseudomilitaris*	BCC 2091	Lepidoptera (larva)	MF416589	MF416535	MF416479	-	MF416441
*Blackwellomyces roseostromatus*	BCC 91358	Lepidoptera (larva)	-	MT003033	MT017845	MT017805	MT017824
*Blackwellomyces roseostromatus*	BCC 91360	Lepidoptera (larva)	-	MT003035	MT017847	MT017807	MT017826
*Blackwellomyces taiwanensis*	NTUPPMCC 20-068	Coleoptera (larva)	-	MT974411	-	-	MW200251
*Blackwellomyces taiwanensis*	NTUPPMCC 20-071	Coleoptera (larva)	-	MT974409	MW200242	MW200246	MW200250
*Cordyceps bifusispora*	EFCC 5690	Lepidoptera (pupa)	EF468952	EF468806	EF468746	EF468854	EF468909
*Cordyceps bifusispora*	EFCC 8260	-	EF468953	EF468807	EF468747	EF468855	EF468910
*Cordyceps coleopterorum*	CBS 110.73	Coleoptera (larva)	JF415965	JF415988	JF416028	JN049903	JF416006
*Cordyceps koratensis*	NHJ 2662	Lepidoptera	-	GQ249982	GQ250032	ON470206	ON470208
*Cordyceps koratensis*	NHJ 666.01	Arachnida	-	GQ249981	GQ250031	-	-
*Cordyceps militaris*	ARSEF 5050	Lepidoptera	-	-	HQ881020	HQ880901	HQ880973
*Cordyceps militaris*	OSC 93623	Lepidoptera (pupa)	AY184977	AY184966	DQ522332	DQ522377	AY545732
*Cordyceps nabanheensis*	YFCC 8409	Lepidoptera (larva)	OL468564	OL468584	OL473532	OL739578	OL473543
*Cordyceps nabanheensis*	YFCC 8410	Lepidoptera (larva)	OL468565	OL468585	OL473533	OL739579	OL473544
*Cordyceps tenuipes*	ARSEF 5135	Lepidoptera (pupa)	MF416612	JF415980	JF416020	JN049896	JF416000
*Corniculantispora aphanocladii*	CBS 797.84	Agaricus bisporus	KM283763	KM283787	KM283811	KM283833	KM283853
*Corniculantispora psalliotae*	CBS 101270	Soil	AF339607	EF469081	EF469066	EF469095	EF469113
*Corniculantispora psalliotae*	CBS 532.81	Soil	AF339609	AF339560	EF469067	EF469096	EF469112
*Engyodontium parvisporum*	IHEM 22910	Human bronchoscopy specimen	-	LC092915	LC425558	-	-
*Engyodontium rectidentatum*	CBS 206.74	Air	-	LC092912	LC425553	-	-
*Engyodontium rectidentatum*	CBS 641.74	Buried keratinous substance	-	LC092914	LC425540	-	-
*Flavocillium bifurcatum*	YFCC 6101	Lepidoptera; Noctuidae	-	MN576781	MN576951	MN576841	MN576897
*Gamszarea humicola*	CGMCC3.19303	Soil	MK311231	MK328997	MK336027	-	MK335979
*Gamszarea wallacei*	CBS 101237	Lepidoptera	AY184978	AY184967	EF469073	EF469102	EF469119
*Gibellula leiopus*	BCC16025	Araneae	MF416602	MF416548	MF416492	MF416649	-
*Gibellula longispora*	NHJ 12014	Araneae	EU369098	-	EU369017	EU369055	EU369075
*Gibellula pulchra*	NHJ 10808	Araneae	EU369099	EU369035	EU369018	EU369056	EU369076
*Hevansia minuta*	BCC 47519	Araneae, *Meotipa* sp.	-	MZ684002	MZ707811	MZ707826	MZ707833
*Hevansia novoguineensis*	CBS 610.80	Araneae	-	MH394646	MH521885	-	MH521844
*Hevansia novoguineensis*	NHJ 11923	Araneae	EU369095	EU369032	EU369013	EU369052	EU369072
*Hevansia novoguineensis*	NHJ 13161	Araneae	EU369093	-	EU369011	EU369050	-
*Jenniferia cinerea*	NHJ 03510	Araneae, *Amyciaea* sp.	EU369091	GQ249970	EU369009	EU369048	EU369070
*Jenniferia thomisidarum*	BCC 37881	Araneae, Diaea cf. dorsata	-	MZ684010	MZ707823	MZ707830	MZ707843
*Jenniferia thomisidarum*	BCC 37882	Araneae, Diaea cf. dorsata	-	MZ684011	MZ707824	MZ707831	MZ707844
*Kanoksria zaquensis*	HMAS 246915	*Ophiocordyceps sinensis*	MT789701	MT789697	MT797812	MT797810	-
*Kanoksria zaquensis*	HMAS 246917	*Ophiocordyceps sinensis*	MT789700	MT789696	MT797811	MT797809	-
*Lecanicillium aranearum*	CBS 726.73a	Arachnida	AF339586	AF339537	EF468781	EF468887	EF468934
*Lecanicillium araneogenum*	GZU1031Lea	Araneae; spider	KX845705	-	KX845697	KX845699	KY845701
*Lecanicillium araneosum*	KY11341	Araneae; spider	-	ON502832	ON525443	-	ON525442
*Lecanicillium araneosum*	KY11342	Araneae; spider	-	ON502837	ON525445	-	ON525444
*Lecanicillium attenuatum*	CBS 402.78	Leaf litter (*Acer saccharum*)	AF339614	AF339565	EF468782	EF468888	EF468935
*Lecanicillium attenuatum*	CBS 170.76	Lepidoptera; Carpocapsa		OP752153	OP762607	OP762611	OP762615
*Lecanicillium attenuatum*	KACC 42493	-	KM283756	KM283780	KM283804	KM283826	KM283846
*Lecanicillium fusisporum*	CBS 164.70	*Coltricia perennis*	AF339598	AF339549	EF468783	EF468889	—
*Lecanicillium lecanii*	CBS 101247	Hemiptera; Coccidae	AF339604	AF339555	DQ522359	DQ522407	DQ522466
*Lecanicillium lepidopterorum*	GZAC SD05151	Lepidoptera (pupa)	-	-	-	-	MT727044
*Lecanicillium longisporum*	CBS 102072	*Macrosiphoniella sanborni* (Hemiptera)	KM283772	KM283796	KM283819	KM283839	KM283861
*Lecanicillium longisporum*	CBS 126.27	Hemiptera; Monophlebidae	KM283773	KM283797	KM283820	KM283840	KM283862
*Lecanicillium muscarium*	CBS 143.62	Hemiptera; Aleyrodidae	KM283774	KM283798	KM283821	KM283841	KM283863
*Lecanicillium neocoleopterorum*	GY11242	Coleoptera; ladybug	-	-	MN097815	MN097817	MN097814
*Lecanicillium neocoleopterorum*	GY11241	Coleoptera; ladybug	-	-	MN097813	MN097816	MN097812
*Lecanicillium pissodis*	BBC7	Heteroptera (*Corythucha arcuata*)	MT004819	MT004835	MT027503	MT027506	MT027509
*Lecanicillium pissodis*	CBS 118231	*Pissodes strobi* adult	KM283775	KM283799	KM283822	KM283842	KM283864
*Lecanicillium sabanense*	ANDES-F 1024	Hemiptera; Coccidae	KC633251	KC875225	KC633266	-	KC633249
*Lecanicillium uredinophilum*	CEP 054	Hemiptera (*Trialeurodes vaporariorum*)	-	OP752150	MH062184	OP762608	OP762612
*Lecanicillium uredinophilum*	CEP 057	Hemiptera (*Trialeurodes vaporariorum*)	-	OP752151	MH062186	OP762609	OP762613
*Lecanicillium uredinophilum*	CEP 108	Hemiptera (*Myzus persicae*)	-	OP752152	MH062188	OP762610	OP762614
*Liangia sinensis*	YFCC3103	Fungi	MN576726	MN576782	MN576952	MN576842	MN576898
*Liangia sinensis*	YFCC3104	Fungi	MN576727	MN576783	MN576953	MN576843	MN576899
** *Liangia sinensis* **	**CGMCC3.25667**	***Ophiocordycepsglobiceps* on Diptera**	**PP177399**	**PP179398**	**PP464672**	**PP464635**	**PP482036**
** *Liangia sinensis* **	**CGMCC3.25668**	***Ophiocordycepsglobiceps* on Diptera**	**PP177400**	**PP179399**	**PP464673**	**PP464636**	**PP482037**
*Neohyperdermium piperis*	CBS 116719	Hemiptera	-	AY466442	DQ118749	DQ127240	EU369083
*Neohyperdermium pulvinatum*	P.C. 602	Hemiptera	-	DQ118738	DQ118746	DQ127237	-
*Neotorrubiella chinghridicola*	BCC 39684	Orthopterida	-	MK632096	MK632071	MK632181	MK632148
*Neotorrubiella chinghridicola*	BCC 80733	Orthopterida	-	MK632097	MK632072	MK632176	MK632149
*Niveomyces albus*	BCC 74477	*Ophiocordyceps* sp. on Arachnida (Araneae)	-	ON103158	ON125016	ON286877	ON125028
*Niveomyces albus*	BCC 83025	*Ophiocordyceps* sp. on Diptera	-	ON103157	ON125015	ON286876	ON125027
*Niveomyces coronatus*	NY 04434800	*Ophiocordyceps camponoti*-*floridani*	ON493547	ON493606	ON513397	ON513399	ON513400
*Niveomyces formicidarum*	BCC 79346	*Ophiocordyceps* sp. on Hymenoptera	-	ON103160	ON125018	ON286878	ON125030
*Niveomyces formicidarum*	BCC 83026	*Ophiocordyceps* sp. on Hymenoptera	-	ON103161	ON125019	ON286879	-
*Niveomyces hirsutellae*	BCC 36631	*Ophiocordyceps* sp. on Hemiptera	-	ON103164	ON125022	ON286882	ON125033
*Niveomyces hirsutellae*	BCC 36632	*Ophiocordyceps* sp. on Hemiptera	-	ON103165	ON125023	ON286883	ON125034
*Niveomyces hirsutellae*	BCC 78482	*Ophiocordyceps* sp. on Hemiptera	-	ON103166	ON125024	ON286884	ON125035
*Niveomyces insectorum*	CBS 756.73	*Palthothyreus tarsatus* in Rubiaceae, associated with *Gibellula formicarum*	-	ON103169	ON125026	ON286887	ON125038
*Niveomyces multisynnematus*	BCC 90307	*Ophiocordyceps* sp. on Hemiptera	-	ON103162	ON125020	ON286880	ON125031
*Niveomyces multisynnematus*	BCC 90308	*Ophiocordyceps* sp. on Hemiptera	-	ON103163	ON125021	ON286881	ON125032
** *Niveomyces multisynnematus* **	**CGMCC3.25670**	***Ophiocordyceps* sp. on Diptera**	**PP177379**	**PP179401**	**PP464674**	**PP464637**	**PP464651**
** *Niveomyces multisynnematus* **	**CGMCC3.25672**	***Ophiocordyceps* sp. on Diptera**	**PP177380**	**PP179403**	**PP464675**	**PP464638**	**PP464652**
*Parengyodontium album*	CBS 368.72	Soil	-	LC092910	LC382183	-	-
*Parengyodontium album*	CBS 504.83	Human brain abscess	-	LC092899	LC382177	-	-
*Pleurodesmospora lepidopterorum*	DY 10501	Lepidoptera	-	-	MW834317	MW834315	MW834316
*Pleurodesmospora lepidopterorum*	DY 10502	Lepidoptera	-	-	MW834319	-	MW834318
*Polystromomyces araneae*	BCC 93301	Arachnida	-	MZ684016	MZ707825	MZ707832	MZ707845
*Pseudogibellula formicarum*	BCC 84257	*Ophiocordyceps flavida*	-	MT512653	MT533480	MT533473	-
*Pseudogibellula formicarum*	CBS 433.73	*Palthothyreus tarsatus*	-	MH872442	MT533481	MT533475	-
*Pseudoniveomyces blattae*	BCC 53567	Blattodea	-	ON103167	-	ON286885	ON125036
*Pseudoniveomyces blattae*	BCC 53568	Blattodea	-	ON103168	ON125025	ON286886	ON125037
*Samsoniella alboaurantium*	CBS 240.32	Lepidoptera (pupa)	JF415958	JF415979	JF416019	JN049895	JF415999
*Samsoniella alboaurantium*	CBS 262.58	Soil	AB080087	MH869308	MF416497	MF416654	MF416448
*Samsoniella alpina*	YFCC 5818	Hepialidae (*Hepialus baimaensis*)	MN576753	MN576809	MN576979	MN576869	MN576923
*Samsoniella alpina*	YFCC 5831	Hepialidae (*Hepialus baimaensis*)	MN576754	MN576810	MN576980	MN576870	MN576924
*Samsoniella anhuiensis*	RCEF2590	Araneae; spider	OR978313	OR978316	OR966516	OR989964	-
*Samsoniella anhuiensis*	RCEF2830	Araneae; spider	OM268843	OM268848	OM483864	OM751889	-
*Samsoniella antleroides*	YFCC 6016	Noctuidae (larvae)	MN576747	MN576803	MN576973	MN576863	MN576917
*Samsoniella antleroides*	YFCC 6113	Noctuidae (larvae)	MN576748	MN576804	MN576974	MN576864	MN576918
*Samsoniella aranea*	RCEF2831	Araneae; spider	OM268844	OM268849	OM483865	OM751882	OM802500
*Samsoniella aranea*	RCEF2868	Araneae; spider	OM268845	OM268850	OM483866	OM751883	OM802501
*Samsoniella aranea*	RCEF2870	Araneae; spider	OR978314	OR978317	OR966517	OR989965	OR989966
*Samsoniella aurantia*	TBRC 7271	Lepidoptera	-	MF140728	MF140846	MF140791	MF140818
*Samsoniella aurantia*	TBRC 7273	Lepidoptera	-	MF140726	MF140844	-	MF140816
*Samsoniella cardinalis*	YFCC 5830	Limacodidae (pupa)	MN576732	MN576788	MN576958	MN576848	MN576902
*Samsoniella cardinalis*	YFCC 6144	Limacodidae (pupa)	MN576730	MN576786	MN576956	MN576846	MN576900
*Samsoniella cardinalis*	YFCC 6320	Limacodidae (pupa)	MN576731	MN576787	MN576957	MN576847	MN576901
*Samsoniella coccinellidicola*	YFCC 8772	Coccinellidae	ON563166	ON621670	ON676514	ON676502	ON568685
*Samsoniella coccinellidicola*	YFCC 8773	Coccinellidae	ON563167	ON621671	ON676515	ON676503	ON568686
*Samsoniella coleopterorum*	A19501	Curculionidae (Snout beetle)	-	-	MN101586	MT642600	MN101585
*Samsoniella coleopterorum*	GY407201	Lepidoptera (larvae)	-	MZ827010	MZ855233	-	MZ855239
*Samsoniella cristata*	YFCC 6023	Saturniidae (pupa)	MN576736	MN576792	MN576962	MN576852	MN576906
*Samsoniella cristata*	YFCC 7004	Saturniidae (pupa)	MN576737	MN576793	MN576963	MN576853	MN576907
*Samsoniella duyunensis*	DY07501	Formicidae (Ant)	-	OR263307	OR282780	OR282773	OR282776
*Samsoniella duyunensis*	DY09161	Lepidoptera (pupa)	-	OQ363112	OQ398145	OR296698	OQ397660
*Samsoniella erucae*	KY11121	Lepidoptera (Caterpillar)	-	ON502835	ON525425	-	ON525424
*Samsoniella erucae*	KY11122	Lepidoptera (Caterpillar)	-	ON502822	ON525427	-	ON525426
*Samsoniella farinospora*	YFCC 8774	Araneae; spider	ON563168	ON621672	ON676516	ON676504	ON568687
*Samsoniella farinospora*	YFCC 9051	Lepidoptera; Hepialus	ON563169	ON621673	ON676517	ON676505	ON568688
*Samsoniella fusiformispora*	RCEF5406	Araneae; spider	OM268846	OM268851	OM483867	OM751890	-
*Samsoniella haniana*	YFCC 8769	Lepidoptera (pupa)	ON563170	ON621674	ON676518	ON676506	ON568689
*Samsoniella haniana*	YFCC 8770	Lepidoptera (pupa)	ON563171	ON621675	ON676519	ON676507	ON568690
*Samsoniella hepiali*	Cor-4	Fungi (*Ophiocordyceps sinensis*)	MN576743	MN576799	MN576969	MN576859	MN576913
*Samsoniella hepiali*	ICMM 82-2	Fungi (*Ophiocordyceps sinensis*)	MN576738	MN576794	MN576964	MN576854	MN576908
*Samsoniella hepiali*	YFCC 7024	Lepidoptera	MN576741	MN576797	MN576967	MN576857	MN576911
*Samsoniella hymenopterorum*	A19521	Vespidae (Bee)	-	-	MN101588	MT642601	MT642604
*Samsoniella hymenopterorum*	A19522	Vespidae (Bee)	-	-	MN101591	MN101589	MN101590
*Samsoniella inthanonensis*	TBRC 7915	Lepidoptera (pupa)	-	MF140725	MF140849	MF140790	MF140815
*Samsoniella kunmingensis*	YHH 16002	Lepidoptera (pupa)	MN576746	MN576802	MN576972	MN576862	MN576916
*Samsoniella lanmaoa*	YFCC 6148	Lepidoptera (pupa)	MN576733	MN576789	MN576959	MN576849	MN576903
*Samsoniella lanmaoa*	YFCC 6193	Lepidoptera (pupa)	MN576734	MN576790	MN576960	MN576850	MN576904
*Samsoniella lasiocampidarum*	NTUPPMCC 20-061	Lepidoptera (larva)	-	MT974364	MW200220	MW200229	-
*Samsoniella lasiocampidarum*	NTUPPMCC 20-062	Lepidoptera (larva)	-	MT974361	MW200218	MW200227	MW200236
*Samsoniella neopupicola*	KY11321	Lepidoptera (pupa)	-	ON502839	ON525433	-	ON525432
*Samsoniella neopupicola*	KY11322	Lepidoptera (pupa)	-	ON502833	ON525435	-	ON525434
*Samsoniella pseudotortricidae*	YFCC 9052	Lepidoptera (pupa)	ON563173	ON621677	ON676521	ON676509	ON568692
*Samsoniella pseudotortricidae*	YFCC 9053	Lepidoptera (pupa)	ON563174	ON621678	ON676522	ON676510	ON568693
*Samsoniella pupicola*	DY101681	Lepidoptera (pupa)	-	MZ827009	MZ855231	-	MZ855237
*Samsoniella pupicola*	DY101682	Lepidoptera (pupa)	-	MZ827635	MZ855232	-	MZ855238
*Samsoniella ramosa*	YFCC 6020	Lepidoptera (pupa)	MN576749	MN576805	MN576975	MN576865	MN576919
** *Samsoniella sanmingense* **	**CGMCC3.25661**	**Lepidoptera (larva)**	**PP177395**	**PP179392**	**PP482033**	**PP464664**	**PP464647**
** *Samsoniella sanmingense* **	**CGMCC3.25662**	**Lepidoptera (larva)**	**PP177396**	**PP179393**	**PP482034**	**PP464665**	**PP464648**
*Samsoniella sinensis*	YFCC 8766	Lepidoptera (larvae)	ON563175	ON621679	ON676523	ON676511	ON568694
*Samsoniella sinensis*	YFCC 8767	Dermaptera	ON563176	ON621680	ON676524	ON676512	ON568695
*Samsoniella tiankengensis*	KY11741	Lepidoptera (pupa)	-	ON502838	ON525437	-	ON525436
*Samsoniella tiankengensis*	KY11742	Lepidoptera (pupa)	-	ON502841	ON525439	-	ON525438
*Samsoniella tortricidae*	YFCC 6013	Lepidoptera (pupa)	MN576751	MN576807	MN576977	MN576867	MN576921
*Samsoniella tortricidae*	YFCC 6131	Lepidoptera (pupa)	MN576750	MN576806	MN576976	MN576866	MN576920
*Samsoniella vallis*	DY07241	Lepidoptera (pupa)	-	OR263306	OR282778	OR282772	OR282774
*Samsoniella vallis*	DY07242	Lepidoptera (pupa)	-	OR263308	OR282779	-	OR282775
*Samsoniella winandae*	TBRC 17511	Lepidoptera (Cocoon)	-	OM491231	OM687896	OM687901	OM687899
*Samsoniella winandae*	TBRC 17512	Lepidoptera (pupa)	-	OM491232	OM687897	OM687902	OM687900
*Samsoniella yuanzuiensis*	NTUPPMCC 20-065	Lepidoptera (pupa)	-	MT974360	MW200217	MW200226	MW200235
*Samsoniella yunnanensis*	YFCC 1527	Fungi (*Cordyceps cicadae*)	MN576756	MN576812	MN576982	MN576872	MN576926
*Samsoniella yunnanensis*	YFCC 1824	Fungi (*Cordyceps cicadae*)	MN576757	MN576813	MN576983	MN576873	MN576927
*Simplicillium lamellicola*	CBS 116.25	Soil	AF339601	AF339552	DQ522356	DQ522404	DQ522462
*Simplicillium lanosoniveum*	CBS 101267	Fungi (*Hemileia vastatrix*)	AF339603	AF339554	DQ522357	DQ522405	DQ522463
*Simplicillium lanosoniveum*	CBS 704.86	Fungi (*Hemileia vastatrix*)	AF339602	AF339553	DQ522358	DQ522406	DQ522464
*Simplicillium obclavatum*	CBS 311.74	Air above sugarcane field	AF339567	AF339517	EF468798	-	-
*Trichoderma deliquescens*	ATCC 208838	On decorticated conifer wood	AF543768	AF543791	AF543781	AY489662	DQ522446
*Trichoderma stercorarium*	ATCC 62321	Cow dung	AF543769	AF543792	AF543782	AY489633	EF469103
*Zouia cauligalbarum*	GZUIFRZHJ01	Lepidoptera (Stemborer)	MH730665	MH730667	MH801920	MH801922	MH801924
*Zouia cauligalbarum*	GZUIFRZHJ02	Lepidoptera (Stemborer)	MH730666	MH730668	MH801921	MH801923	MH801925

Note: Newly generated sequences are in bold.

**Table 3 jof-11-00028-t003:** Bayesian estimates of divergence times (Mya), including 95% posterior density (HPD) and Bayesian posterior probabilities (BYPP), for the major 33 nodes in Figure 2.

Node	Mean (Mya)/95% HPD (Mya)/BYPP	Node	Mean (Mya)/95% HPD (Mya)/BYPP
1	133.64/125.06–142.88/1.00	18	48.44/44.71–52.18/1.00
2	103.80/94.70–112.91/1.00	19	46.32/42.74–50.13/-
3	101.99/100.99–102.95/1.00	20	45.95/33.86–58.83/1.00
4	95.25/90.35–99.62/-	21	44.90/39.67–50.59/0.99
5	88.91/83.36–95.03/1	22	44.24/42.74–50.13/-
6	81.44/76.23–86.67/1.00	23	43.17/38.55–47.96/1.00
7	77.42/72.36–82.89/-	24	43.07/39.90–46.44/0.95
8	75.82/70.02–81.86/-	25	40.98/36.32–45.92/1.00
9	73.34/66.20–80.29/-	26	39.56/36.33–42.85/1.00
10	70.81/64.51–77.00/1.00	27	34.57/31.41–37.67/1.00
11	63.57/58.85–68.69/-	28	34.11/25.41–42.93/-
12	61.85/56.28–67.78/1	29	29.32/26.68–31.97/1.00
13	60.78/55.31–66.35/1	30	26.70/21.42–31.96/-
14	59.24/54.54–64.03/-	31	22.46/20.04–24.96/1.00
15	57.53/51.65–63.41/-	32	20.50/18.40–22.81/0.93
16	52.26/47.14–57.94/1	33	16.43/12.76–20.03/0.97
17	48.64/42.96–54.84/-		

## Data Availability

All sequences generated in this study were submitted to the NCBIdatabase.

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
