# Peer review of "Molecular Phylogenetics and Estimation of Evolutionary Divergence and Biogeography of the Family Cordycipitaceae (Ascomycota, Hypocreales)"

_jof, 2025, doi:10.3390/jof11010028_

Round 1

Reviewer 1 Report

The Cordycipitaceae family of insecticidal fungi is the most complex in the order Hypocreales (Ascomycota), displaying a variety of morphological characteristics and diverse host ranges and are widely distributed in nature. Data provide a significant addition to the diversity, ecology, and evolutionary aspects of the Cordycipitaceae.

There is no line numbering from page 1 to page 23.

 Keywords repeat the Manuscript title. This is not help indexers and search engines find relevant papers. Keywords should represent the content of your manuscript and be specific to your field or sub-field.

Examples: Manuscript title: Direct observation of nonlinear optics in an isolated carbon nanotube

Poor keywords: molecule, optics, lasers, energy lifetime

Better keywords: single-molecule interaction, Kerr effect, carbon nanotubes, energy level structure

https://www.springer.com/gp/authors-editors/authorandreviewertutorials/writing-a-journal-manuscript/title-abstract-and-keywords/10285522

Page 5. 2.1. Sampling, Fungal Isolation, and Morphology. The phrase «The growing edge of fungal mycelium was transferred to a new PDA plate and this step was repeated until pure cultures were obtained» is not needed. This is an absolutely ordinary process of obtaining a pure culture.

SEM Microscopy is not very high quality. The form of conidia is broken. Preliminary fixation is required.  Figure 4. v, w; Figure 5. h, o

Page 16. lines 29   Figure 5.  (l–p) -- It must be (l–o)

 There are no «Conclusions» at the end of the manuscript.

Author Response

Dear Editors and Reviewers:

Thank you for your letter and comments relating to our manuscript entitled “Molecular Phylogenetic, Taxonomic Classification, and Biogeography of the Family Cordycipitaceae (Ascomycota, Hypocreales)” (ID: jof-3383591). The comments were very helpful in revising and improving our manuscript as well as emphasizingthe significance to our research. We have read the comments carefully and made corrections accordingly. Revised portions are marked in blue in the manuscript. The main corrections in the paper and our responses to the reviewer’s comments are given below. We hope that the revisions in the manuscript and our accompanying responses will be sufficient to make our manuscript suitable for publication in the Journal of Fungi.

Responses to the comments of the reviewer:

Reviewer#1

Comments 1: The Cordycipitaceae family of insecticidal fungi is the most complex in the order Hypocreales (Ascomycota), displaying a variety of morphological characteristics and diverse host ranges and are widely distributed in nature. Data provide a significant addition to the diversity, ecology, and evolutionary aspects of the Cordycipitaceae.

Response 1: We have revised it.

Comments 2: There is no line numbering from page 1 to page 23.

Response 2: We have revised it.

Comments 3: Keywords repeat the Manuscript title. This is not help indexers and search engines find relevant papers. Keywords should represent the content of your manuscript and be specific to your field or sub-field.

Examples: Manuscript title: Direct observation of nonlinear optics in an isolated carbon nanotube

Poor keywords: molecule, optics, lasers, energy lifetime

Better keywords: single-molecule interaction, Kerr effect, carbon nanotubes, energy level structure

Response 3: We have revised it.

Keywords: Ancestral state reconstruction; molecular clock; morphology; multilocus phylogeny; taxonomy

Comments 4: Page 5. 2.1. Sampling, Fungal Isolation, and Morphology. The phrase «The growing edge of fungal mycelium was transferred to a new PDA plate and this step was repeated until pure cultures were obtained» is not needed. This is an absolutely ordinary process of obtaining a pure culture.

Response 4: We have revised it.

Comments 5: SEM Microscopy is not very high quality. The form of conidia is broken. Preliminary fixation is required.  Figure 4. v, w; Figure 5. h, o Page 16. lines 29   Figure 5.  (l–p) -- It must be (l–o).

Response 5: We have revised them and provided more detailed SEM images of the conidial structures.

Comments 6: There are no «Conclusions» at the end of the manuscript.

Response 6: We have added Conclusions at the end of the manuscript.

Conclusions

In this study, based on the combination of morphological observations and multi-locus phylogenetic analyses, we identified two new species, two new record species and one known species of Cordycipitaceae distributed within five different genera from China, viz. Akanthomyces baishanensis sp. nov., Samsoniella sanmingense sp. nov., Blackwellomyces lateris, Niveomyces multisynnematus and Liangia sinensis. Molecular clock analyses examining divergence times, along with historical biogeographic analyses, suggest that the ancestor of Akanthomyces sensulato (s.l.) most likely originated in Asia during the Paleogene period, approximately 34.57 Mya. These findings provide valuable help for the classification, diversity, and evolutionary aspects of Cordycipitaceae species.

We tried our best to improve the manuscript and made some changes marked in blue in revised paper which will not influence the content and framework of the paper. We appreciate for Editors/Reviewers’ warm work earnestly and hope the revision will meet with your approval. Once again, thank you very much for your comments and suggestions.

Kind regards,

Junzhi Qiu

E-mail address: junzhiqiu@126.com

Reviewer 2 Report

According to the title the main objective is related to phylogeny and biogeography of Cordycipitaceae, please modify the abstract in this way. New records and new species resulted in a secondary part of this work.

Also the title indicate taxonomic classification, but this part is not included in the paper, I mean it is not a new proposal inside Cordycipitaceae family, there are new records and new species. Please remove taxonomic classification in the title.

Page 3 lines 18- 31. This paragraph corresponds to a mix of materials and methods and results, please write as objectives.

Page 6, lines 44- 50 the beginning of the discussion actually are results, modify please as a results.

Page 2, line 1, misspelling in word “Cordycepitaceae”

Page 2, line 17, misspelling in word “Cordycepitaceae”

Page 2 correct the name “Cordycepitaceae” in all names of this page

Page 3 lines 18- 31. This paragraph corresponds to a mix of materials and methods and results, please write as objectives.

Page 5, line 7, please consult what is the correct way to cite elongation factor 1 alfa, consult GenBank

Page 5, line 11, verify the name of the primer LROR, I think that is LR0R, zero instead of O

Page 5, line 39, about analyses with Bayesian Inference, I am not sure that the sampling frequency was 2,000,000 generations, this are the number of MCMC? Or indicate wich ones were the MCMC number.

Page 7, line33, Why only tef-1 was compared?

Author Response

Dear Editors and Reviewers:

Thank you for your letter and comments relating to our manuscript entitled “Molecular Phylogenetic, Taxonomic Classification, and Biogeography of the Family Cordycipitaceae (Ascomycota, Hypocreales)” (ID: jof-3383591). The comments were very helpful in revising and improving our manuscript as well as emphasizingthe significance to our research. We have read the comments carefully and made corrections accordingly. Revised portions are marked in blue in the manuscript. The main corrections in the paper and our responses to the reviewer’s comments are given below. We hope that the revisions in the manuscript and our accompanying responses will be sufficient to make our manuscript suitable for publication in the Journal of Fungi.

Responses to the comments of the reviewer:

Reviewer#2

Comments 1: According to the title the main objective is related to phylogeny and biogeography of Cordycipitaceae, please modify the abstract in this way. New records and new species resulted in a secondary part of this work.

Response 1: We have revised it.

Abstract: The Cordycipitaceae family of insecticidal fungi is the most complex in the order Hypocreales (Ascomycota), displaying a variety of morphological characteristics and diverse host ranges and are widely distributed in nature. Data provide a significant addition to the diversity, ecology, and evolutionary aspects of the Cordycipitaceae. Based on Bayesian evolutionary analysis of five genomic loci: the small subunit ribosomal RNA (SSU) gene, the large subunit of ribosomal RNA (LSU) gene, the translation elongation factor 1-α (tef1-α) gene, the largest subunit of RNA polymerase II (rpb1) and the second largest subunit of RNA polymerase II (rpb2), we inferred the divergence times for members of the Cordycipitaceae, improving the internal phylogeny of this fungal family. Molecular clock analyses indicate that the ancestor of Akanthomyces sensu lato occurred in the Paleogene period (34.57 Mya, 95% HPD: 31.41–37.67 Mya), and that most species appeared in the Neogene period. The historical biogeography of Akanthomyces sensu lato was reconstructed using reconstructing ancestral state in phylogenies (RASP) analyses, indicating that it most likely originated in Asia. Combined morphological characterization and phylogenetic analyses were used to identify and taxonomic place five species within Cordycipitaceae. These include: (i) two new species, namely Akanthomyces baishanensis sp. nov. and Samsoniella sanmingense sp. nov., (ii) a new record species isolated from infected from Lepidopteran host, Blackwellomyces lateris, (iii) a new record species in the genus Niveomyces, with sporothrix-like asexual morph, namely N. multisynnematus, isolated from dipteran insects (flies), and (iv) a known species of the (hyper-) mycoparasite, Liangia sinensis, isolated from the fungus Ophiocordyceps globiceps (Ophiocordycipitaceae) growing on a dipteran host.

Comments 2: Also the title indicate taxonomic classification, but this part is not included in the paper, I mean it is not a new proposal inside Cordycipitaceae family, there are new records and new species. Please remove taxonomic classification in the title.

Response 2: We have revised it.

Molecular phylogenetic and estimation of evolutionary divergence and biogeography of the Family Cordycipitaceae (Ascomycota, Hypocreales)

Comments 3: Page 3 lines 18- 31. This paragraph corresponds to a mix of materials and methods and results, please write as objectives.

Response 3: We have revised it.

Most entomopathogenic fungi belong to Cordycipitaceae of Hypocreales, however, the species diversity, the taxonomic status and phylogenetic relation-ships of species within Cordycipitaceae have not been fully clarified. During an exploration aimed at delineating the diversity and geographical distribution of entomopathogenic fungi in China, a series of intriguing specimens were collected in Fujian and Jilin Province, China. These isolates were characterized via molecular sequencing of five genomic loci, and phylogenetic analysis was used to clarify their taxonomic relationships within Cordycipitaceae. These data were combined with morphological characterization, and we report the description of two new species, two new record species and one known species, which are illustrated and detailed described. Concurrently, we utilized Paleoophiocordyceps coccophagus G. H. Sung, Poinar & Spatafora as a fossil calibration for molecular clock analysis to infer the divergence times of members of Cordycipitaceae, increasing our understanding of the phylogeny, origin and evolutionary history of this family.

Comments 4: Page 6, lines 44- 50 the beginning of the discussion actually are results, modify please as a results.

Response 4: We have changed it.

  1. Results

3.1. Phylogenetic analyses

Phylogenetic analyses and morphological comparisons presented herein indicate that the Fujian and Jilin Province specimens collected belong to Cordycipitaceae and group within the Liangia, Niveomyces and Samsoniella genera. Two new species (Akanthomyces baishanensis sp. nov. and Samsoniella sanmingense sp. nov.), two new record species (Blackwellomyces lateris and Niveomyces multisynnematus) and one known species (Liangia sinensis) were described and their phylogenetic placements were determined (Table 2, Figure 1).

Phylogenetic analyses were based on the concatenated nucleotide sequence dataset derived from the SSU, LSU, tef–1α, rpb1, and rpb2 gene loci, and were examined across 29 taxa representing 235 sequences within Cordycipitaceae, and which included Trichoderma (T. deliquescens ATCC 208838 and T. stercorarium ATCC 62321) as the outgroup (Table 2, Figure 1). The combined dataset consisted of 4473 characters (i.e., SSU: 1–1002 bp, LSU: 1003-1734 bp, tef1-α: 1735-2768 bp, rpb1: 2769-3455 bp, rpb2: 3456-4473 bp), in which 2593 characters were constant, 2450 distinct patterns, and 355 singleton sites and 1525 parsimony–informative were identified. The best substitution model for BI analysis included SYM + I + G4 for SSU, SYM + G4 for rpb2 and GTR+F+I+G4 for LSU + tef1-α + rpb1, resulting in an average standard deviation of split frequencies = 0.007254. For ML and BI phylogenetic analyses, the inferred topologies were largely consistent across most branches, with Ak. baishanensis, B. lateris, L. sinensis, N. multisynnematus and S. sanmingense found to be distributed within distinct clades with strong support.

Comments 5: Page 2, line 1, misspelling in word “Cordycepitaceae”

Response 5: We have revised it.

Comments 6: Page 2, line 17, misspelling in word “Cordycepitaceae”

Response 6: We have revised it.

Comments 7: Page 2 correct the name “Cordycepitaceae” in all names of this page

Response 7: We have revised it and rechecked the spelling of "Cordycipitaceae" throughout the manuscript.

Comments 8: Page 3 lines 18- 31. This paragraph corresponds to a mix of materials and methods and results, please write as objectives.

Response 8: We have revised it.

Comments 9: Page 5, line 7, please consult what is the correct way to cite elongation factor 1 alfa, consult GenBank

Response 9: We have revised it.

EF1α-EF: GCTCCYGGHCAYCGTGAYTTYAT

EF1α-ER: ATGACACCRACRGCRACRGTYTG

Comments 10: Page 5, line 11, verify the name of the primer LROR, I think that is LR0R, zero instead of O

Response 10: We have changed “LROR” to “LR0R”.

Comments 11: Page 5, line 39, about analyses with Bayesian Inference, I am not sure that the sampling frequency was 2,000,000 generations, this are the number of MCMC? Or indicate wich ones were the MCMC number.

Response 11: We have revised it.

BI analysis was performed with MrBayes 3.2.6, and the best evolutionary model was decided under the Akaike Information Criterion (AIC) via PartitionFinder2, in which the first 1/4 of the trees were discarded as burn-in trees and four parallel Markov Chain Monte Carlo chains sampled every 100 steps for 2,000,000 generations until the average standard deviation reaching 0.01.

Comments 12: Page 7, line33, Why only tef-1 was compared?

Response 12: We have revised it.

According to the studies by Wang et al. [1], species delimitations by the tree topologies for the individual loci and the genetic divergence comparisons showed that the tef1-α sequence data provided the best resolution distinguishing Samsoniella spp., followed by rpb1 sequences. Due to the lack of rpb1 sequence in S. tiankengensis, we only compared the tef1-α sequence. By comparing the tef1-α sequences of the two species, we were able to get a clearer picture of the genetic differences between them. The comparison of nucleotide sequences showed that there are 2.6% differences (0.8% in rpb2, 1.8% bp in tef1-α) between S. tiankengensis and S. pupicola, suggesting they are separated species [2-3]. In this study, the comparison of nucleotide sequences showed that there are 2.3% differences (0.6% in rpb2, 1.7% bp in tef1-α) between S. sanmingense and S. pupicola.

  1. Wang, Y.; Wang, Z.Q.; Thanarut, C.; Dao, V.M.; Wsng, Y.B.; Yu, H. Phylogeny and species delimitations in the economically, medically, and ecologically important genus Samsoniella (Cordycipitaceae, Hypocreales). MycoKeys 2023, 99, 227–250. Doi 10.3897/mycokeys.99.106474
  2. Chen, W.H.; Liang, J.D.; Ren, X.X.; Zhao, J.H.; Han, F.Y.; Liang, Z.Q. Cryptic diversity of Isaria-like species in Guizhou, China. Life 2021, 11, 1093. Doi 10.3390/life11101093
  3. Chen, W.H.; Liang, J.D.; Ren, X.X.; Zhao, J.H.; Han, F.Y.; Liang, Z.Q. Species diversity of Cordyceps–like fungi in the Tiankeng karst region of China. Microbiol. Spectrum 2022, 5, e0197522. Doi 10.1128/spectrum.01975-22.

We tried our best to improve the manuscript and made some changes marked in blue in revised paper which will not influence the content and framework of the paper. We appreciate for Editors/Reviewers’ warm work earnestly and hope the revision will meet with your approval. Once again, thank you very much for your comments and suggestions.

Kind regards,

Junzhi Qiu

E-mail address: junzhiqiu@126.com
